# A Curriculum Perspective to Robust Loss Functions

## Abstract

Learning with noisy labels is a fundamental problem in machine learning. Much work has been done in designing loss functions that are theoretically robust against label noise. However, it remains unclear why robust loss functions can underfit and why loss functions deviating from theoretical robustness conditions can appear robust. To elucidate these questions, we show that most robust loss functions differ only in the sample-weighting curriculums they implicitly define. The curriculum perspective enables straightforward analysis of the training dynamics with each loss function, which has not been considered in existing theoretical approaches. We show that underfitting can be attributed to marginal sample weights during training, and noise robustness can be attributed to larger weights for clean samples than noisy samples. With a simple fix to the curriculums, robust loss functions that severely underfit can become competitive with the state-of-the-art.[1]

## 1 Introduction

Labeling errors are non-negligible from automatic annotation (Liu et al., 2021; Khayrallah & Koehn, 2018), crowd-sourcing (Russakovsky et al., 2015) and expert annotation (Kato & Matsubara, 2010; Bridge et al., 2016). The resulting noisy labels may hamper generalization since over-parameterized neural networks can memorize the training set (Zhang et al., 2017). To combat the adverse impact of noisy labels in classification tasks, a large body of research (Song et al., 2020) aims to design loss functions robust against label noise. Most existing approaches derive sufficient conditions (Ghosh et al., 2017; Zhou et al., 2021b) for noise robustness. Despite the theoretical appeal being agnostic to models and training dynamics[2], they may fail to comprehensively characterize the performance of robust loss functions. Specifically, it has been shown that (1) robust loss functions can underfit difficult tasks (Zhang & Sabuncu, 2018; Wang et al., 2019c; Ma et al., 2020), while (2) loss functions violating existing robustness conditions (Zhang & Sabuncu, 2018; Wang et al., 2019c;b) can exhibit robustness. For (1), existing explanations (Ma et al., 2020; Wang et al., 2019a) can be limited as discussed in §2.2. For (2), to our knowledge, there has been no work directly addressing it.

We analyze training dynamics with various loss functions to elucidate the above observations, which complements existing theoretical approaches. Specifically, we rewrite a broad array of loss functions into a standard form with the same implicit loss function and varied sample-weighting functions (§3), each implicitly defining a sample-weighting curriculum. The interaction between the sample-weighting function and the distribution of implicit losses of samples thus reveals aspects of the training dynamics with each loss function. Here a curriculum by definition (Wang et al., 2020) specifies a sequence of re-weighting for the distribution of training samples, e.g., sample weighting (Chang et al., 2017) or sample selection (Zhou et al., 2021a), based on a metric for sample difficulty. Notably, our novel curriculum perspective first connects robust loss functions to the seemingly distinct curriculum learning approaches (Song et al., 2020) for noise-robust training.

With our curriculum perspective, we first attribute the underfitting issue of robust loss functions to marginal sample weights during training (§4.1). In particular, for classification tasks with numerous classes, the initial sample weights under the curriculum of some robust loss functions can become marginal. When modifying the curriculums accordingly, robust loss functions that severely underfit can become competitive with the state-of-the-art. We then attribute noise robustness of loss functions to larger sample weights for clean samples than for noisy ones during training (§4.2). By examining

---

[1]Our code will be available at `github`.
[2]Changes of model states during training except for trivial metrics like evaluation metrics and loss functions.

the changes of implicit losses during training, we find that dynamics of SGD suppress the learning of noisy samples. Curriculums of robust loss functions further suppress the learning of noisy samples by magnifying the difference in learning pace between clean and noisy samples while neglecting unlearned noisy samples. Based on our analysis, we present two unexpected phenomenons when viewed from existing theoretical results. By simply changing the learning rate schedule, (1) robust loss functions *can* become vulnerable to label noise, while (2) cross entropy *can* appear robust.

## 2 BACKGROUND

**Classification** $k$-ary classification with input $\boldsymbol{x} \in \mathbb{R}^d$ can be solved by classifier $\arg\max_i s_i$, where $s_i$ is the score for the $i$-th class in the class scoring function $\boldsymbol{s} : \mathbb{R}^d \to \mathbb{R}^k$ parameterized by $\boldsymbol{\theta}$. Class scores $\boldsymbol{s}(\boldsymbol{x}; \boldsymbol{\theta})$ can be turned into class probabilities with the softmax function $p_i = e^{s_i}/(\sum_{j=1}^k e^{s_j})$, where $p_i$ is the probability of class $i$. Given a loss function $L(\boldsymbol{s}(\boldsymbol{x}; \boldsymbol{\theta}), y)$ and data $(\boldsymbol{x}, y)$ with $y \in \{1..k\}$ the ground truth label, $\boldsymbol{\theta}$ can be estimated by risk minimization $\arg\min_{\boldsymbol{\theta}} \mathbb{E}_{\boldsymbol{x},y} L(\boldsymbol{s}(\boldsymbol{x}; \boldsymbol{\theta}), y)$, whose solutions are called risk minimizers. We use $\boldsymbol{s}$ in place of $\boldsymbol{s}(\boldsymbol{x}; \boldsymbol{\theta})$ for notation simplicity.

**Noise robustness** Mistakes in the labeling process can corrupt the clean label $y$ into a noisy label

$$\tilde{y} = \begin{cases} y, & \text{with probability } P(\tilde{y} = y|\boldsymbol{x}, y) \\ i, \ i \neq y & \text{with probability } P(\tilde{y} = i|\boldsymbol{x}, y) \end{cases}$$

Label noise is symmetric (or uniform) if $P(\tilde{y} = i|\boldsymbol{x}, y) = \eta/(k-1), \forall i \neq y$, with $\eta = P(\tilde{y} \neq y)$ the noise rate constant. Label noise is asymmetric (or class-conditional) if $P(\tilde{y} = i|\boldsymbol{x}, y) = P(\tilde{y} = i|y)$. Given data $(\boldsymbol{x}, \tilde{y})$ with noisy label $\tilde{y}$, a loss function $L$ is robust against label noise if

$$\arg\min_{\boldsymbol{\theta}} \mathbb{E}_{\boldsymbol{x},\tilde{y}} L(\boldsymbol{s}(\boldsymbol{x}; \boldsymbol{\theta}), \tilde{y}) = \arg\min_{\boldsymbol{\theta}} \mathbb{E}_{\boldsymbol{x},y} L(\boldsymbol{s}(\boldsymbol{x}; \boldsymbol{\theta}), y) \tag{1}$$

**Conditions for noise robustness** Most existing approaches on robust loss function (Ghosh et al., 2017; Ma et al., 2020; Liu & Guo, 2020; Feng et al., 2020; Zhou et al., 2021b) focus on bounding the difference between risk minimizers obtained with noisy and clean data, i.e., ensuring that Eq. (1) approximately holds. These bounds only depend on the loss functions and mild assumptions about the dataset. To contrast our curriculum perspective with these approaches, we review two typical sufficient conditions for noise robustness. Loss function $L$ is symmetric (Ghosh et al., 2017) if

$$\sum_{i=1}^k L(\boldsymbol{s}, i) = C, \ \forall \boldsymbol{s} \in \mathbb{R}^k, \tag{2}$$

where $C$ is a constant. It is robust against *symmetric* label noise with $\eta < (k-1)/k$. This stringent condition was later relaxed to the asymmetric condition. To rephrase, a loss function as a function of the softmax probability $p_i$, i.e., $L(\boldsymbol{s}, i) = l(p_i)$, is asymmetric (Zhou et al., 2021b) if

$$\max_{i \neq y} \frac{P(\tilde{y} = i|\boldsymbol{x}, y)}{P(\tilde{y} = y|\boldsymbol{x}, y)} = \tilde{r} \leq r = \inf_{\substack{0 \leq p_i, \Delta p \leq 1 \\ p_i + \Delta p \leq 1}} \frac{l(p_i) - l(p_i + \Delta p)}{l(0) - l(\Delta p)}, \tag{3}$$

where $\Delta p$ is a valid increment of $p_i$. When clean labels dominate the data, i.e., $\tilde{r} < 1$, an asymmetric loss is robust against *generic* label noise.

**The active-passive dichotomy** Ma et al. (2020) draw a distinction between active and passive loss functions. By rewriting loss function $L$ into a sum of basic functions, $L(\boldsymbol{s}, y) = \sum_{i=1}^k l(\boldsymbol{s}, i)$, active loss functions can be defined with $\forall i \neq y, \ l(\boldsymbol{s}, i) = 0$, which emphasizes learning the target label. In contrast, passive loss functions defined with $\exists i \neq y, \ l(\boldsymbol{s}, i) \neq 0$ can be improved by unlearning the non-target labels. However, since there is no canonical guideline to specify $l(\boldsymbol{s}, i)$, different specifications can lead to ambiguities in the active-passive dichotomy as discussed in §2.2.

In summary, the above research fails to address open questions in §2.2. Since many loss functions degrade less in performance than cross entropy under label noise, exhibiting various degrees of noise robustness, as a slight abuse of terminology, we refer to them as robust loss functions hereafter.

### 2.1 TYPICAL ROBUST LOSS FUNCTIONS

We review typical robust loss functions for our analysis besides cross entropy (CE) that is vulnerable to label noise (Ghosh et al., 2017). Differences in constant scaling factors and additive biases are ignored, as they are either equivalent to learning rate scaling in SGD or irrelevant in the gradient computation. See Table 1 for the formulas and Appendix A for an extended review of loss functions.

**Symmetric** The mean absolute error (MAE; Ghosh et al., 2017) and the equivalent reverse cross entropy (RCE; Wang et al., 2019c) are both symmetric as they satisfy Eq. (2). Ma et al. (2020) make loss functions satisfying $L(\boldsymbol{s}, i) > 0, \forall i \in \{1..k\}$ symmetric by normalizing them with $L_{\mathrm{N}}(\boldsymbol{s}, y) = L(\boldsymbol{s}, y)/(\sum_{i=1}^{k} L(\boldsymbol{s}, i))$. We include normalized cross entropy (NCE; Ma et al. 2020) as an example.

**Asymmetric** We include asymmetric generalized cross entropy (AGCE) and asymmetric unhinged loss (AUL) proposed by Zhou et al. (2021b) as typical asymmetric loss functions. Notably, AGCE with $q \geq 1$ and AUL with $q \leq 1$ are completely asymmetric, i.e., Eq. (3) always holds when $\tilde{r} < 1$.

**Combined** Loss functions can be combined for both robustness and sufficient learning. For example, generalized cross entropy (GCE; Zhang & Sabuncu, 2018) is a smooth interpolation between CE and MAE. Alternatively, symmetric cross entropy (SCE; Wang et al., 2019c) is a weighted average of CE and RCE (MAE). Ma et al. (2020) argue that robust and sufficient training requires a balanced combination of active and passive loss functions. Accordingly to their active-passive dichotomy, CE and NCE are active while MAE (RCE) is passive. We include NCE+MAE as an example.

## 2.2 OPEN QUESTIONS

**Why do robust loss functions underfit?** Ma et al. (2020) attribute underfitting to failure in balancing active-passive components. However, the active-passive dichotomy can be ambiguous. Given

$$L_{\mathrm{MAE}}(\boldsymbol{s}, y) \propto \sum_{i=1}^{k} |\mathbb{I}(i = y) - p_i| \propto \sum_{i=1}^{k} \mathbb{I}(i = y)(1 - p_i)$$

where $\mathbb{I}(\cdot)$ is the indicator function, MAE is passive with $l(\boldsymbol{s}, i) = |\mathbb{I}(i = y) - p_i|$ but active with $l(\boldsymbol{s}, i) = \mathbb{I}(i = y)(1 - p_i)$. Wang et al. (2019a) view $\|\nabla_{\boldsymbol{s}} L(\boldsymbol{s}, y)\|_1$ as weights for sample gradients and attribute underfitting to their low variance, making clean and noisy samples less distinguishable. However, as we show in §4.1, MAE also underfits data with clean labels. In summary, neither Ma et al. (2020) nor Wang et al. (2019a) fully explain the underfitting issue.

**What affects the robustness of a loss function?** Although combined loss functions such as GCE and SCE fail to satisfy Eq. (2) and (3), it is unclear why they exhibit robustness against label noise (Zhang & Sabuncu, 2018; Wang et al., 2019c). Furthermore, it is unclear how training dynamics with loss functions, which are irrelevant in theoretical robustness guarantees (Ghosh et al., 2017; Zhou et al., 2021b; Ma et al., 2020; Liu & Guo, 2020; Feng et al., 2020), affect their noise robustness.

## 3 IMPLICIT CURRICULUMS OF LOSS FUNCTIONS

Loss functions in Table 1 except NCE and NCE+MAE can be written as a function of the target softmax probability $p_y$, i.e., $L(\boldsymbol{s}, y) = l(p_y)$. A close examination of $p_y$ gives

$$p_y = \frac{e^{s_y}}{\sum_{i=1}^{k} e^{s_i}} = \frac{1}{e^{\log \sum_{i \neq y} e^{s_i - s_y}} + 1} = \frac{1}{e^{-\Delta_y} + 1} = \mathrm{sigmoid}(\Delta_y), \tag{4}$$

where

$$\Delta_y = s_y - \log \sum_{i \neq y} e^{s_i} \leq s_y - \max_{i \neq y} s_i \tag{5}$$

indicates how well a sample is learned, as $\Delta_y \geq 0$ ensures successful classification $y = \arg \max_i s_i$. Loss functions with the form $L(\boldsymbol{s}, y) = l(p_y)$ can thus be rewritten into a standard form with *equivalent gradients*, i.e.,

$$L(\boldsymbol{s}, y) = l(p_y) = \int_{\boldsymbol{s}} \nabla_{\boldsymbol{s}} l(p_y) \mathrm{d}\boldsymbol{s} = \int_{\boldsymbol{s}} l'(p_y) p_y'(\Delta_y) \cdot \nabla_{\boldsymbol{s}} \Delta_y \mathrm{d}\boldsymbol{s} = -w(\Delta_y) \cdot \Delta_y, \tag{6}$$

where $w(\Delta_y) = |l'(p_y) p_y'(\Delta_y)|$ is a scalar weight wrapped with the stop-gradient operator, and $\Delta_y$ is an implicit loss function embedded in $L(\boldsymbol{s}, y)$. Loss functions following Eq. (6) thus implicitly define different sample-weighting curriculums, with $w(\Delta_y)$ the sample-weighting function and $\Delta_y$ the metric for sample difficulty. Notably, $\Delta_y$ factors out the preference from $w(\Delta_y)$, making it a more direct metric for sample difficulty than those based on losses (Kumar et al., 2010) or gradient magnitudes (Gopal, 2016). In addition, the interaction between $w(\Delta_y)$ and $\Delta_y$ distributions reveals aspects of the training dynamics with each loss function. See Table 1 for a summary of $w(\Delta_y)$ for the reviewed loss functions, and Appendix A for how hyperparameters affect $w(\Delta_y)$.

| Type | Name | Function | Sample Weight $w$ | Constraints |
|------|------|----------|-------------------|-------------|
| / | CE | $-\log p_y$ | $1 - p_y$ | / |
| Sym. | MAE/RCE | $1 - p_y$ | $p_y(1 - p_y)$ | / |
| | NCE | $-\log p_y / \left( \sum_{i=1}^{k} -\log p_i \right)$ | / | / |
| Asym. | AUL | $[(a - p_y)^q - (a-1)^q]/q$ | $p_y(1 - p_y)(a - p_y)^{q-1}$ | $a > 1, q > 0$ |
| | AGCE | $[(a+1) - (a + p_y)^q]/q$ | $p_y(a + p_y)^{q-1}(1 - p_y)$ | $a > 0, q > 0$ |
| Comb. | GCE | $(1 - p_y^q)/q$ | $p_y^q(1 - p_y)$ | $0 < q \leq 1$ |
| | SCE | $(1-q) \cdot L_{\mathrm{CE}} + q \cdot L_{\mathrm{MAE}}$ | $(1 - q + q \cdot p_y)(1 - p_y)$ | $0 < q < 1$ |
| | NCE+MAE | $(1-q) \cdot L_{\mathrm{NCE}} + q \cdot L_{\mathrm{MAE}}$ | / | $0 < q < 1$ |

Table 1: Expressions, constraints and sample-weighting functions (§3) for loss functions in §2.1.

### 3.1 THE ADDITIONAL REGULARIZER OF NCE

NCE does not follow Eq. (6) as it additionally depends on $p_i$, $i \neq y$. Yet it can be rewritten into

$$L_{\mathrm{NCE}}(\boldsymbol{s}, y) = \gamma_{\mathrm{NCE}} \cdot L_{\mathrm{CE}}(\boldsymbol{s}, y) + \gamma_{\mathrm{NCE}} \cdot \epsilon_{\mathrm{NCE}} \cdot R_{\mathrm{NCE}}(\boldsymbol{s}), \qquad (7)$$

where $\gamma_{\mathrm{NCE}} = 1/(\sum_{i=1}^{k} -\log p_i)$ and $\epsilon_{\mathrm{NCE}} = k(-\log p_y)/(\sum_{i=1}^{k} -\log p_i)$ are scalar weights wrapped with the stop-gradient operator. In Eq. (7), the first term is a *primary loss function* defining a sample-weighting curriculum similar to CE. The second is a *regularizer*

$$R_{\mathrm{NCE}}(\boldsymbol{s}) = \sum_{i=1}^{k} \frac{1}{k} \log p_i$$

reducing the entropy of the softmax output. Although the training dynamics of NCE are complicated by the additional regularizer, we can use the upperbound for the L1 norm of gradient $\nabla_{\boldsymbol{s}} L_{\mathrm{NCE}}(\boldsymbol{s}, y)$,

$$\hat{w}_{\mathrm{NCE}} = 2\gamma_{\mathrm{NCE}} \cdot w_{\mathrm{CE}}\left(1 + \epsilon_{\mathrm{NCE}}\right) \geq \|\nabla_{\boldsymbol{s}} L_{\mathrm{NCE}}(\boldsymbol{s}, y)\|_1, \qquad (8)$$

as the weight of each sample in parameter updates. Examining how $\hat{w}_{\mathrm{NCE}}$ changes during training helps understand why NCE underfits in §4.1. We leave derivations of Eq. (7) and (8) and discussions on similar loss functions with an additional regularizer to Appendix A.5.

## 4 ROBUST LOSS FUNCTIONS FROM THE CURRICULUM PERSPECTIVE

We examine the interaction between $w(\Delta_y)$ and $\Delta_y$ distributions to address questions in §2.2. Results are reported on MNIST (Lecun et al., 1998) and CIFAR10/100 (Krizhevsky, 2009) with synthetic symmetric and asymmetric label noise following Ma et al. (2020); Zhou et al. (2021b). For real-world scenarios, we include CIFAR10/100 with human label noise (Wei et al., 2022) and the large-scale noisy dataset WebVision (Li et al., 2017), which exhibit more complex noise patterns than symmetric and asymmetric label noise. Unlike standard settings, we scale $w(\Delta_y)$ to unit maximum to avoid complications, since hyperparameters of loss functions can change the scale of $w(\Delta_y)$, essentially adjusting the learning rate of SGD. See Appendix B for more experimental details.

### 4.1 UNDERSTANDING UNDERFITTING OF ROBUST LOSS FUNCTIONS

**Robust loss functions can underfit** We confirm that on difficult tasks like CIFAR100 (Song et al., 2020), underfitting can result from robust loss functions themselves rather than inferior hyperparameters. As shown in Table 2, CE outperforms NCE, AGCE and AUL by a nontrivial margin on CIFAR100. In particular, MAE performs much worse than CE, similar to AGCE[†] and AUL[†] with inferior hyperparameters. See Table 9 for similar results with learning rate $\alpha = 0.01$.

**Marginal[3] sample weights explains underfitting** The effective scale of parameter update at step $t$ during SGD can be approximated by $\alpha_t^* = \alpha_t \cdot \bar{w}_t$, where $\bar{w}_t$ is the average sample weight of the

---

[3]Very small or too small to be important.

| Underfitting | Loss | CIFAR100 Acc. | $\bar{\alpha}_t^*$ | CIFAR10 Acc. | $\bar{\alpha}_t^*$ |
|---|---|---|---|---|---|
| / | CE | $71.33 \pm 0.23$ | 8.183 | $92.76 \pm 0.30$ | 5.541 |
| No | SCE | $71.36 \pm 0.39$ | 9.541 | $93.17 \pm 0.06$ | 7.018 |
| | GCE | $69.95 \pm 0.40$ | 8.861 | $92.96 \pm 0.13$ | 6.151 |
| | NCE+MAE | $68.89 \pm 0.23$ | / | $92.37 \pm 0.33$ | / |
| Moderate | AUL | $58.75 \pm 1.07$ | 5.278 | $92.43 \pm 0.19$ | 5.171 |
| | AGCE | $49.27 \pm 1.03$ | 4.537 | $92.61 \pm 0.18$ | 5.225 |
| | NCE | $43.18 \pm 1.55$ | / | $91.28 \pm 0.22$ | / |
| Severe | MAE | $3.69 \pm 0.59$ | 0.035 | $91.56 \pm 0.11$ | 2.492 |
| | AUL$^\dagger$ | $3.13 \pm 0.43$ | 0.033 | $91.13 \pm 0.06$ | 2.308 |
| | AGCE$^\dagger$ | $1.62 \pm 0.69$ | 0.009 | $87.14 \pm 4.96$ | 1.701 |

Table 2: Without label noise, robust loss functions can underfit CIFAR100 but CIFAR10. Hyperparameters of loss functions are tuned on CIFAR100 and listed in Table 8. We report test accuracy and $\bar{\alpha}_t^*$ (scaled by $10^3$) at the final training step from 3 different runs with learning rate $\alpha = 0.1$. Loss functions with inferior hyperparameters (denoted with $\dagger$) are included as references. See Table 9 for similar results with learning rate $\alpha = 0.01$.

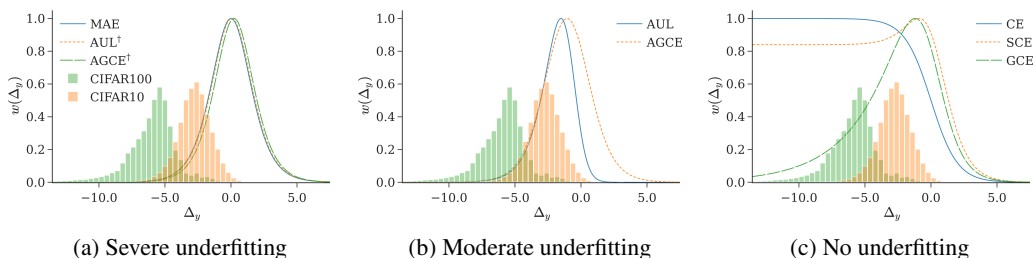

(a) Severe underfitting      (b) Moderate underfitting      (c) No underfitting

Figure 1: Sample-weighting functions $w(\Delta_y)$ of loss functions in Table 2 with hyperparameters in Table 8. We include the initial $\Delta_y$ distributions of CIFAR10 and CIFAR100 for reference, which are obtained by computing $\Delta_y$ with a randomly initialized model for all training samples.

batch and $\alpha_t$ the learning rate. The overall $\alpha_t^*$ up to step $t$ can be $\bar{\alpha}_t^* = \sum_{i=1}^{t} \alpha_i^* / t$. In Table 2, for loss functions that heavily underfit on CIFAR100, $\bar{\alpha}_t^*$ at the final step is marginal compared to CE, suggesting a marginal overall sample weight during training given the same learning rate schedule.

**Underfitting from fast diminishing sample weights** Similar to CE, in Fig. 2a, $\bar{\alpha}_t^*$ of NCE based on $\hat{w}_{\text{NCE}}$ peaks at initialization. However, it decreases much faster than CE since both $\gamma_{\text{NCE}}$ and $w_{\text{CE}}$ decrease with improved $\Delta_y$. In addition, the regularizer $R_{\text{NCE}}(s)$ further reduces the entropy of softmax output and thus $\gamma_{\text{NCE}}$. The resulting fast decreasing $\hat{w}_{\text{NCE}}$ hampers the learning of training samples, which can lead to underfitting.

**Underfitting from marginal initial sample weights** In Fig. 1a, unlike NCE, loss functions that severely underfit in Table 2 assign marginal weights to samples in CIFAR100 at initialization, which leads to marginal initial $\bar{\alpha}_t^*$. $\Delta_y$ of these samples can barely improve before the learning rate vanishes, thus leading to underfitting. In contrast, loss functions with non-trivial initial sample weights (Fig. 1b and 1c) result in moderate or no underfitting. As further corroboration, we plot $\bar{\alpha}_t^*$ of AUL with superior and inferior hyperparameters (AUL and AUL$^\dagger$ in Table 2) in Fig. 2b. $\bar{\alpha}_t^*$ stays marginal with AUL$^\dagger$, but quickly increases to a non-negligible value before gradually decreasing with AUL.

**Loss combination can mitigate underfitting.** As $\hat{w}_{\text{NCE}}$ peaks at initialization but quickly diminishes while $w_{\text{MAE}}$ is marginal at initialization but peaks later during training, combining NCE with MAE can mitigate the underfitting issue of each other. In Table 2, combining NCE and MAE suffers less from underfitting compared to both individuals.

**Increased number of classes leads to marginal initial sample weights.** Unlike CIFAR100, all loss functions in Table 2 perform equally well on CIFAR10. Such a difference has been vaguely attributed to the increased task difficulty of CIFAR100 (Zhang & Sabuncu, 2018; Song et al., 2020). Intuitively, the more classes, the more subtle differences to be distinguished. In addition, the number of classes $k$ determines the initial distribution of $\Delta_y$. Assume that class scores $s_i$ at *initialization* are i.i.d. normal variables $s_i \sim \mathcal{N}(\mu, \sigma)$. In particular, $\mu = 0$ and $\sigma = 1$ for most neural networks with standard initializations (Glorot & Bengio, 2010; He et al., 2015) and normalization layers (Ioffe &

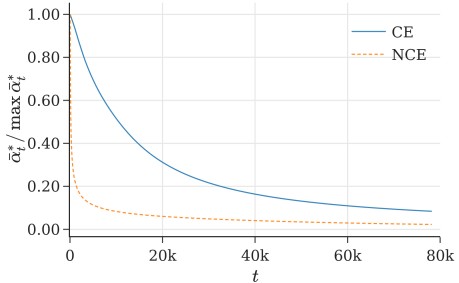
(a) NCE with estimated weight upperbound.

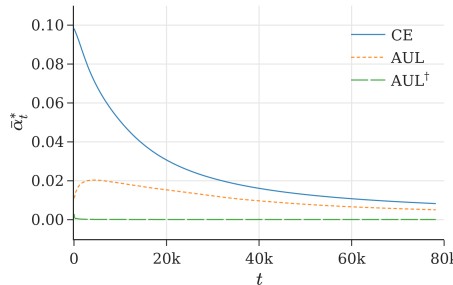
(b) AUL with inferior/superior hyperparameters.

Figure 2: Different explanations for underfitting: (a) fast diminishing sample weights; (b) marginal initial sample weights. We plot the variation of $\bar{\alpha}_t^*$ with training step $t$ on CIFAR100 without label noise for each loss function. $\bar{\alpha}_t^*$ of NCE is estimated with $\hat{w}_{\mathrm{NCE}}$. Since $\hat{w}_{\mathrm{NCE}}$ is not comparable to $w_{\mathrm{CE}}$, we normalize $\bar{\alpha}_t^*$ with its maximum in (a) to emphasize its variation during training.

| Loss | Clean $\eta = 0$ | Symmetric $\eta = 0.4$ | $\eta = 0.8$ | Asymmetric $\eta = 0.4$ | Human $\eta = 0.4$ |
|---|---|---|---|---|---|
| CE[‡] | $71.33 \pm 0.43$ | $39.92 \pm 0.10$ | $7.59 \pm 0.20$ | $40.17 \pm 1.31$ | / |
| GCE[‡] | $63.09 \pm 1.39$ | $56.11 \pm 1.35$ | $17.42 \pm 0.06$ | $40.91 \pm 0.57$ | / |
| NCE[‡] | $29.96 \pm 0.73$ | $19.54 \pm 0.52$ | $8.55 \pm 0.37$ | $20.64 \pm 0.40$ | / |
| NCE+AUL[‡] | $68.96 \pm 0.16$ | $59.25 \pm 0.23$ | $23.03 \pm 0.64$ | $38.59 \pm 0.48$ | / |
| AGCE | $49.27 \pm 1.03$ | $47.76 \pm 1.75$ | $16.03 \pm 0.59$ | $33.40 \pm 1.57$ | $30.45 \pm 1.50$ |
| AGCE shift | $69.39 \pm 0.84$ | $48.21 \pm 1.06$ | $14.49 \pm 0.17$ | $40.76 \pm 0.74$ | $48.71 \pm 0.45$ |
| AGCE scale | $70.57 \pm 0.62$ | $56.69 \pm 0.33$ | $14.64 \pm 0.79$ | $39.71 \pm 0.17$ | $50.85 \pm 0.11$ |
| MAE | $3.69 \pm 0.59$ | $1.29 \pm 0.50$ | $1.00 \pm 0.00$ | $2.53 \pm 1.34$ | $2.09 \pm 0.55$ |
| MAE shift | $68.57 \pm 0.54$ | $49.95 \pm 0.16$ | $13.10 \pm 0.41$ | $39.83 \pm 0.18$ | $47.91 \pm 0.36$ |
| MAE scale | $\mathbf{70.97 \pm 0.41}$ | $\mathbf{60.57 \pm 1.04}$ | $\mathbf{24.44 \pm 0.73}$ | $\mathbf{44.48 \pm 1.05}$ | $\mathbf{54.70 \pm 0.48}$ |

Table 3: Shifting or scaling $w(\Delta_y)$ mitigates underfitting on CIFAR100 under different label noise. We report test accuracies with 3 different runs. Results from Zhou et al. (2021b) are included as context (denoted with ‡). See Table 10 for hyperparameter $\tau$ of each setting, and Tables 11 and 12 for results with more noise rates.

Szegedy, 2015; Ba et al., 2016). The expected $\Delta_y$ can be approximated with

$$\mathbb{E}[\Delta_y] \approx -\log(k-1) - \sigma^2/2 + \frac{e^{\sigma^2} - 1}{2(k-1)} \tag{9}$$

We leave derivations and comparisons between our assumptions and real settings to Appendix C.1. A large $k$ results in small initial $\Delta_y$; with sample-weighting functions in Fig. 1a it further leads to marginal initial sample weights, which results in underfitting on CIFAR100 as discussed previously.

### 4.1.1 Addressing Underfitting from Marginal Initial Sample Weights

Our analysis suggests that the fixed sample-weighting function $w(\Delta_y)$ is to blame for underfitting. To make the initial sample weights agnostic to the number of classes, we can simply scale

$$w^*(\Delta_y) = w(\Delta_y^*) = w(\Delta_y/|\mathbb{E}[\Delta_y]| \cdot \tau)$$

or shift

$$w^+(\Delta_y) = w(\Delta_y^+) = w(\Delta_y + |\mathbb{E}[\Delta_y]| - \tau)$$

the sample-weighting functions, where $\tau$ is a hyperparameter. Intuitively, $|\mathbb{E}[\Delta_y]|$ in $w^*(\Delta_y)$ and $w^+(\Delta_y)$ cancels the effect of $k$ on the weight of the expected initial $\Delta_y$. A small $\tau$ thus leads to high initial sample weights regardless of $k$. In Appendix C.1.1 we visualize $w^*_{\mathrm{MAE}}(\Delta_y)$ and $w^+_{\mathrm{MAE}}(\Delta_y)$ in Fig. 7 and discuss the robustness of the loss functions they induce.

Results on CIFAR100 with different label noise are reported in Table 3. See Tables 11 and 12 in Appendix C.1.1 for results with additional noise rates. We also report results on the large-scale WebVision dataset with different numbers of classes in Table 4. In summary, shifting and scaling alleviate underfitting, making MAE and AGCE comparable to the previous state-of-the-art (NCE+AUL; Zhou et al. 2021b). Notably, $w^*(\Delta_y)$ leads to dramatic improvements for MAE under all settings.

| Settings | $k = 50$ $\tau = 2.0$ | $k = 200$ $\tau = 1.8$ | $k = 400$ $\tau = 1.6$ |
|---|---|---|---|
| CE | 66.40 | 70.26 | 70.16 |
| MAE | 3.68 | 0.50 | 0.25 |
| MAE shift | 60.76 | 59.31 | 47.32 |
| MAE scale | **66.72** | **71.92** | **71.87** |

Table 4: Shifting or scaling $w(\Delta_y)$ mitigates underfitting on WebVision subsampled with different numbers of classes. $k = 50$ is the standard "mini" setting in previous work (Ma et al., 2020; Zhou et al., 2021b). We report test accuracy with a single run due to a limited computation budget.

| | Clean | Asymmetric | | Symmetric | | | | | | Human | |
|---|---|---|---|---|---|---|---|---|---|---|---|
| | | $\eta = 0.2$ | | $\eta = 0.2$ | | $\eta = 0.4$ | | $\eta = 0.8$ | | $\eta = 0.4$ | |
| Loss | Acc | diff | snr | diff | snr | diff | snr | diff | snr | diff | snr |
| CE | 90.64 | -7.06 | 0.32 | -15.47 | 0.39 | -31.95 | 0.57 | -50.87 | 0.77 | -28.51 | 0.53 |
| SCE | 89.87 | -5.39 | 0.51 | -3.84 | 0.99 | -10.47 | 1.27 | -27.25 | 1.51 | -15.84 | 0.86 |
| GCE | 90.44 | -7.42 | 0.36 | -6.80 | 0.96 | -23.23 | 0.89 | -45.32 | 1.04 | -21.94 | 0.72 |
| AUL | 89.90 | -2.51 | 0.81 | -2.07 | 3.10 | -5.87 | 2.96 | -13.90 | 2.83 | -12.08 | 1.11 |
| MAE | 89.29 | -2.21 | 1.00 | -1.92 | 3.56 | -4.36 | 3.33 | -11.53 | 3.22 | -10.35 | 1.32 |
| AGCE | 82.62 | -9.42 | 0.92 | -1.55 | 3.02 | -19.90 | 2.16 | -41.11 | 1.83 | -21.73 | 1.28 |

Table 5: Robust loss functions assign larger weights to clean samples. We report snr and diff from the best of 5 runs on CIFAR10 under each noise setting, as inferior initialization can heavily degrade the performance. Hyperparameters listed in Table 13 are selected to cover more variants of sample-weighting functions (plotted in Fig. 8), which are not necessarily optimal.

Although $w^*(\Delta_y)$ and $w^+(\Delta_y)$ are agnostic to the number of classes at initialization, their performances differ significantly. Intuitively, $w^+(\Delta_y)$ diminishes much faster than $w^*(\Delta_y)$ with increased $\Delta_y$, which can lead to insufficient training of clean samples and thus inferior performance.

## 4.2 UNDERSTANDING NOISE ROBUSTNESS OF LOSS FUNCTIONS

We show that robust loss functions following Eq. (6) *implicitly* assign larger weights to clean samples. The underlying reasons are explored by examining how $\Delta_y$ distributions change during training. Notably, similar sample-weighting rules are *explicitly* adopted by curriculums for noise robust training (Ren et al., 2018). We leave NCE to future work as it involves an additional regularizer.

**Robust loss functions assign larger weights to clean samples.** We use the ratio between the average weights of clean ($\bar{w}_{\text{clean}}$) and noisy ($\bar{w}_{\text{noise}}$) samples, $\text{snr} = \bar{w}_{\text{clean}}/\bar{w}_{\text{noise}}$, to characterize their relative contribution during training. See Appendix C.2 for the exact formulas. Noise robustness is characterized by differences in test accuracy compared to results with clean labels (diff). We report diff and snr under different label noise on CIFAR10 in Table 5. Loss functions with higher snr have less performance drop with label noise in general, thus being more robust.

To explain what leads to a large snr, we plot changes of $\Delta_y$ distributions during training on CIFAR10 with symmetric label noise in Fig. 3. See Fig. 9 and 10 for similar results with additional types of label noise and loss functions. When trained with loss functions that are more robust against label noise (Fig. 3b and 3c), $\Delta_y$ distributions of noisy and clean samples spread wider and get better separated. In addition, the consistent decrease of $\Delta_y$ for noisy samples suggests that they can be *unlearned*. In contrast, training with CE (Fig. 3a) results in more compact and less separated $\Delta_y$ distributions. Furthermore, $\Delta_y$ of noisy samples consistently increases.

**Dynamics of SGD suppress learning of noisy samples.** As shown in Fig. 3a, noisy samples are learned slower than clean samples as measured by improvements of $\Delta_y$, which can be explained by more coherent gradients among clean samples (Chatterjee & Zielinski, 2022). Similar results have been reported (Zhang et al., 2017; Arpit et al., 2017) and utilized in curriculum-based robust training (Yao et al., 2019; Han et al., 2018). In addition, noisy samples can be unlearned as shown in Fig. 3b and 3c, which can stem from generalization with clean samples. Both dynamics suppress the learning of noisy samples but clean ones, thus leading to robustness against label noise.

**Robust $w(\Delta_y)$ synergizes with SGD dynamics for noise robustness.** In Fig. 1, the bell-shaped $w(\Delta_y)$ of robust loss functions only assigns large weights to samples with moderate $\Delta_y$. Since $\Delta_y$ distributions initially concentrate at the monotonically increasing interval of $w(\Delta_y)$, (1) samples

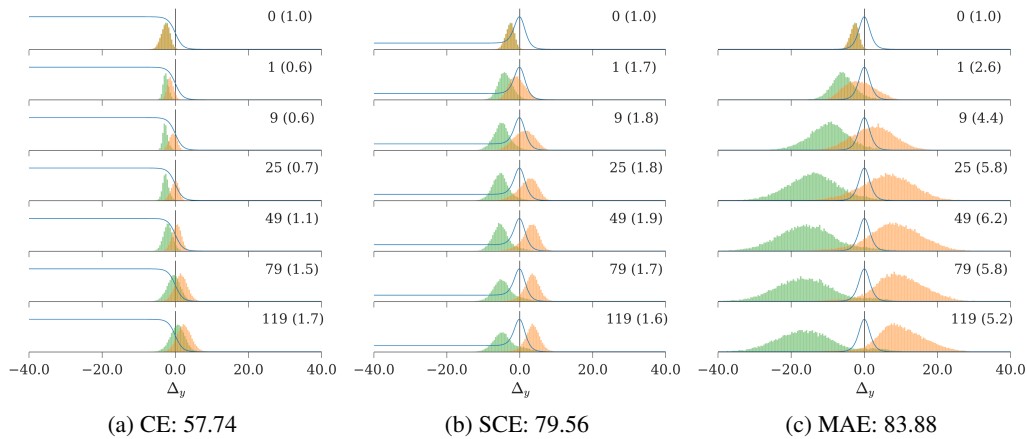

(a) CE: 57.74      (b) SCE: 79.56      (c) MAE: 83.88

Figure 3: How $\Delta_y$ distributions of noisy (green, left) and clean (orange, right) samples change on CIFAR10 during training with symmetric label noise and $\eta = 0.4$. Vertical axes denoting probability density are scaled to the peak of histograms for readability, with epoch number (axis scaling factor) denoted on the right of each subplot. We plot $w(\Delta_y)$ and report the test accuracy of each setting for reference. See Appendix C.2 for results with additional types of label noise and loss functions.

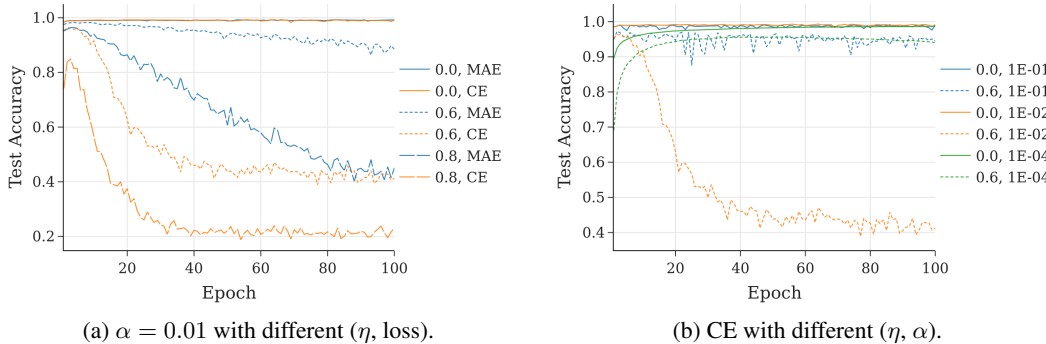

(a) $\alpha = 0.01$ with different $(\eta, \text{loss})$.      (b) CE with different $(\eta, \alpha)$.

Figure 4: Learning curves with fixed learning rate and extended training epochs on MNIST under symmetric label noise, where $\alpha$ is the learning rate and $\eta$ the noise rate.

with faster improving $\Delta_y$, due to either larger initial weights or faster learning as clean samples, are weighted more during early training and learned faster. The magnified learning pace difference explains the widely spread distributions in Fig. 3b and 3c. In addition, (2) the unlearned samples with small $\Delta_y$ receive diminishing weights from $w(\Delta_y)$, which hampers their pace of learning. Noisy samples in Fig. 3b and 3c are consistently unlearned and ignored with marginal sample weights, leading to a consistent decrease in $\Delta_y$. In addition to the SGD dynamics, (1) and (2) further suppress the learning of noisy samples and enhance that of clean samples, thus leading to increased robustness against label noise. In contrast, the monotonically decreasing $w_{\text{CE}}(\Delta_y)$ emphasizes samples with smaller $\Delta_y$, essentially acting against the SGD dynamics for noise robustness. Thus training with CE results in increased vulnerability to label noise as shown in Table 5.

### 4.2.1 TRAINING SCHEDULES AFFECT NOISE ROBUSTNESS

Although the learning pace of noisy samples gets initially suppressed, the expected gradient will eventually be dominated by noisy samples, since well-learned clean samples receive marginal sample weights thanks to the monotonically decreasing interval of $w(\Delta_y)$. Models with extended training[4] thus risk overfitting noisy samples during the late training stage. Adjusting the training schedules to enable or avoid such overfitting can therefore affect the noise robustness of models. Based on this intuition, we present two phenomenons on noise robustness as *examples* that follow our curriculum perspective but deviate from existing theoretical results:

**Extended training can make robust loss functions vulnerable to label noise.** The learning curves of CE and MAE with *constant* learning rates on MNIST are shown in Fig. 4a. Despite the theoreti-

---

[4]Enough training steps without early stopping or diminishing learning rates for a small training loss.

cally guaranteed noise robustness (Ghosh et al., 2017), similar to CE, with extended training, MAE eventually overfits noisy samples, resulting in vulnerability to label noise.

**CE can become robust by adjusting the learning rate schedule.** To avoid overfitting noisy samples, we can avoid learning when noisy samples dominate the expected gradient. It can be achieved with either early stopping (Song et al., 2019), or a constrained learning pace that prevents sufficient learning of clean samples, which avoids diminishing weights for them. We show the learning curve of CE using fixed learning rates under symmetric label noise on MNIST in Fig. 4b. By simply increasing or decreasing the learning rate, which strengthens the implicit regularization of SGD (Smith et al., 2021) or directly slows down the learning pace, CE can become robust against label noise.

## 5 RELATED WORK

Our work closely relates to robust loss functions against label noise (Song et al., 2020). Most existing studies (Ghosh et al., 2017; Zhang & Sabuncu, 2018; Wang et al., 2019c; Ma et al., 2020; Liu & Guo, 2020; Cheng et al., 2021; Feng et al., 2020; Zhou et al., 2021b) focus on bounding the difference between risk minimizers obtained with noisy and clean data, which are agnostic to training dynamics. In contrast, with our novel curriculum perspective, we analyze the training dynamics with robust loss functions for reasons behind their underfitting issue and noise robustness. The underfitting problem has been heuristically mitigated with loss combination (Zhang & Sabuncu, 2018; Wang et al., 2019c; Ma et al., 2020). We identify the cause and provide effective solutions.

Curriculum-based approaches combat label noise with either sample selection (Chen et al., 2019; Zhou et al., 2021a) or sample weighting (Chang et al., 2017; Jiang et al., 2018; Ren et al., 2018). In particular, sample weights are *explicitly* designed (Wang et al., 2019a;b; Chang et al., 2017) or predicted by a model trained on a different dataset (Jiang et al., 2018; Ren et al., 2018). In contrast, the sample weights in this work are *implicitly* defined by robust loss functions. Notably, the implicit loss function we identified is a more direct metric for sample difficulty compared to common metrics based on loss functions (Kumar et al., 2010; Loshchilov & Hutter, 2015) and gradient magnitudes (Gopal, 2016), which are implicitly affected by the preference from the sample-weighting functions of loss functions. Our work is also related to the ongoing debate (Hacohen & Weinshall, 2019; Wang et al., 2020) on strategies for selecting or weighting samples in curriculum learning: either easier first (Bengio et al., 2009; Kumar et al., 2010) or harder first (Loshchilov & Hutter, 2015; Zhang et al., 2018). The implicit curriculums of robust loss functions can be viewed as a combination of both strategies, emphasizing samples with moderate difficulty.

Most related to our work, Wang et al. (2019b) identify gradient norms as weights for sample gradients and propose heuristic designs of weighting functions for noise-robust training. In contrast, we explicitly identify the implicit loss function, which connects robust loss functions to curriculum learning, facilitates our analysis of the training dynamics and helps elucidate the robustness of loss functions from a curriculum perspective.

Altering noise robustness by adjusting the learning rate is reminiscent of (Huang et al., 2019). They use a cyclic learning rate to make models change back and forth between overfitting and underfitting to collect statistics for noisy label detection. To achieve noise robustness, they discard samples with detected noisy labels and retrain the model from scratch. In contrast, our results show that simply changing the learning rate can achieve noise robustness.

## 6 CONCLUSION AND DISCUSSION

We identified the implicit sample-weighting curriculums of a broad array of loss functions. Our novel curriculum perspective enables examining the training dynamics with loss functions through the interaction between the sample-weighting function and distributions of implicit losses. It connects robust loss functions to the seemingly distinct curriculum learning. Notably, the implicit loss function we identified is a direct metric for sample difficulty in curriculum learning as it factors out the preference of sample-weighting functions. We elucidate the reasons behind underfitting and robustness against label noise and propose a simple approach to address the underfitting issue.

As with previous work on robust loss functions, our empirical results are based on image classification using convolutional neural networks with and without residual connections. We have extended our experiments to cover larger-scale classification tasks, human label noise and a broader array of robust loss functions. Although our derivation does not depend on the models and task specifications, additional experiments should be performed in future work to extend our conclusions to more models and tasks.

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

## A  EXTENDED REVIEW OF LOSS FUNCTIONS

Due to limited space, we only briefly describe typical robust loss functions in §2.1. As a general reference, here we provide a comprehensive review of loss functions related to the standard form Eq. (6). Similar to §2.1, we ignore the differences in constant scaling factors and additive bias. Loss functions and their sample-weighting functions are summarized in Table 6. We examine how hyperparameters affect their sample-weighting functions in Fig. 5.

### A.1  LOSS FUNCTIONS WITHOUT ROBUSTNESS GUARANTEES

**Cross Entropy** (CE)
$$L_{\mathrm{CE}}(\boldsymbol{s}, y) = -\log p_y$$
is the standard loss function for classification.

**Focal Loss** (FL; Lin et al. 2017)
$$L_{\mathrm{FL}}(\boldsymbol{s}, y) = -(1 - p_y)^q \log p_y$$
aims to address label imbalance when training object detection models. Both CE and FL are neither symmetric (Ma et al., 2020) nor asymmetric (Zhou et al., 2021b).

### A.2  SYMMETRIC LOSS FUNCTIONS

**Mean Absolute Error** (MAE; Ghosh et al. 2017)

$$L_{\mathrm{MAE}}(\boldsymbol{s}, y) = \sum_{i=1}^{k} |\mathbb{I}(i = y) - p_i| = 2 - 2p_y \propto 1 - p_y$$

is a classic symmetric loss function, where $\mathbb{I}(i = y)$ is the indicator function.

**Reverse Cross Entropy** (RCE; Wang et al. 2019c)

$$L_{\mathrm{RCE}}(\boldsymbol{s}, y) = \sum_{i=1}^{k} p_i \log \mathbf{1}(i = y) = \sum_{i \neq y} p_i A = (1 - p_y)A \propto 1 - p_y = L_{\mathrm{MAE}}(\boldsymbol{s}, y)$$

is equivalent to MAE in implementation, where $\log 0$ is truncated to a negative constant $A$ to avoid numerical overflow.

Ma et al. (2020) argued that any generic loss functions with $L(\boldsymbol{s}, i) > 0, \forall i \in \{1..k\}$ can become symmetric by simply normalizing them. As an example,

| Name | Function | Sample Weight $w$ | Constraints |
|---|---|---|---|
| CE | $-\log p_y$ | $1 - p_y$ | |
| FL | $-(1-p_y)^q \log p_y$ | $(1-p_y)^q(1-p_y-qp_y\log p_y)$ | $q > 0$ |
| MAE/RCE | $1 - p_y$ | $p_y(1-p_y)$ | |
| AUL | $\frac{(a+1)-(a+p_y)^q}{q}$ | $p_y(1-p_y)(a-p_y)^{q-1}$ | $a>1, q>0$ |
| AGCE | $\frac{(a-p_y)^q-(a-1)^q}{q}$ | $p_y(a+p_y)^{q-1}(1-p_y)$ | $a>0, q>0$ |
| AEL | $e^{-p_y/q}$ | $\frac{1}{q}p_y(1-p_y)e^{-p_y/q}$ | $q>0$ |
| GCE | $(1-p_y^q)/q$ | $p_y^q(1-p_y)$ | $0<q\leq 1$ |
| SCE | $-(1-q)\log p_y + q(1-p_y)$ | $(1-q+q\cdot p_y)(1-p_y)$ | $0<q<1$ |
| TCE | $\sum_{i=1}^{q}(1-p_y)^i/i$ | $p_y\sum_{i=1}^{q}(1-p_y)^i$ | $q\geq 1$ |

Table 6: Expressions, constraints of hyperparameters and sample-weighting functions of loss functions reviewed in Appendix A that follow the standard form Eq. (6).

**Normalized Cross Entropy** (NCE; Ma et al. 2020)

$$L_{\text{NCE}}(\boldsymbol{s}, y) = \frac{L_{\text{CE}}(\boldsymbol{s}, y)}{\sum_{i=1}^{k} L_{\text{CE}}(\boldsymbol{s}, i)} = \frac{-\log p_y}{\sum_{i=1}^{k} -\log p_i}$$

is a symmetric loss function. However, NCE does not follow the standard form of Eq. (6) as it additionally depends on $p_i$, $i \neq y$. It involves an additional regularizer, thus being more relevant to discussions in Appendix A.5.

### A.3 ASYMMETRIC LOSS FUNCTIONS

Zhou et al. (2021b) derived the asymmetric condition for noise robustness and propose numerous asymmetric loss functions:
**Asymmetric Generalized Cross Entropy** (AGCE)

$$L_{\text{AGCE}}(\boldsymbol{s}, y) = \frac{(a+1)-(a+p_y)^q}{q}$$

where $a > 0$ and $q > 0$. It is asymmetric when $\mathbb{I}(q \leq 1)(\frac{a+1}{a})^{1-q} + \mathbb{I}(q > 1) \leq 1/\tilde{r}$.
**Asymmetric Unhinged Loss** (AUL)

$$L_{\text{AUL}}(\boldsymbol{s}, y) = \frac{(a-p_y)^q-(a-1)^q}{q}$$

where $a > 1$ and $q > 0$. It is asymmetric when $\mathbb{I}(q \leq 1)(\frac{a}{a-1})^{q-1} + \mathbb{I}(q \leq 1) \leq 1/\tilde{r}$.
**Asymmetric Exponential Loss** (AEL)

$$L_{\text{AEL}}(\boldsymbol{s}, y) = e^{-p_y/q}$$

where $q > 0$. It is asymmetric when $e^{1/q} \leq 1/\tilde{r}$.

### A.4 COMBINED LOSS FUNCTIONS

**Generalized Cross Entropy** (GCE; Zhang & Sabuncu 2018)

$$L_{\text{GCE}}(\boldsymbol{s}, y) = \frac{1-p_y^q}{q}$$

can be viewed as a smooth interpolation between CE and MAE, where $0 < q \leq 1$. CE or MAE can be recovered by setting $q \to 0$ or $q = 1$.

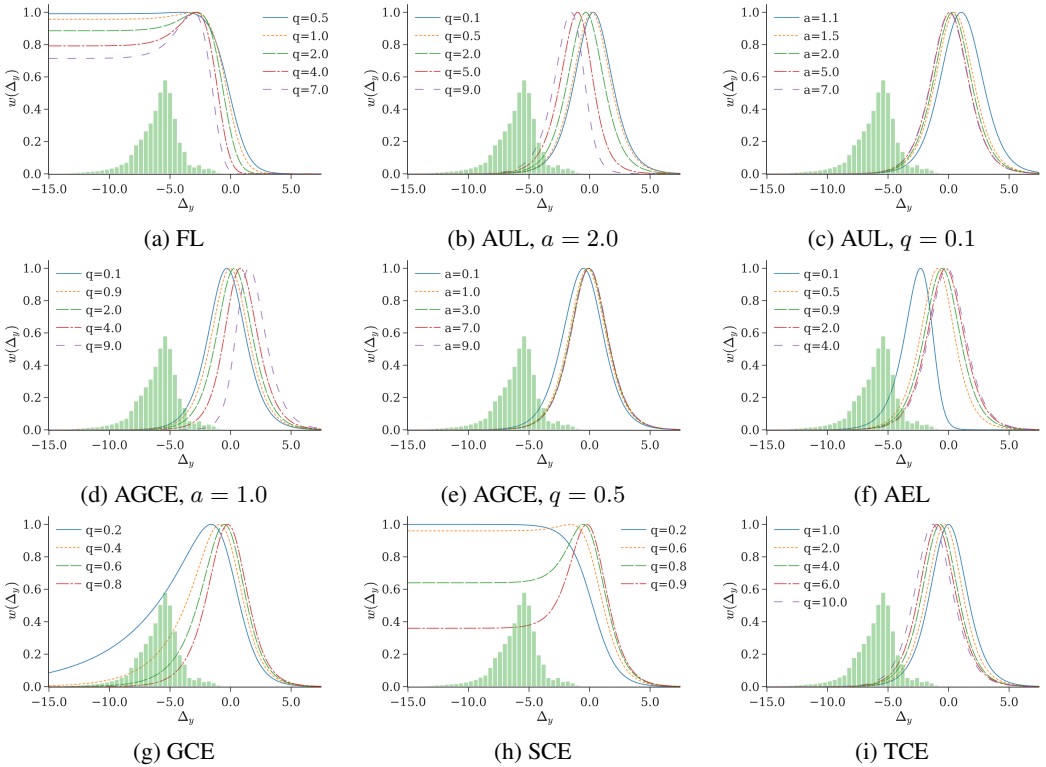

Figure 5: How hyperparameters affect the sample-weighting functions in Table 6. The initial $\Delta_y$ distributions of CIFAR100 extracted with a randomly initialized model are included as reference.

**Symmetric Cross Entropy** (SCE; Wang et al. 2019c)

$$L_{\text{SCE}}(\boldsymbol{s}, y) = a \cdot L_{\text{CE}}(\boldsymbol{s}, y) + b \cdot L_{\text{RCE}}(\boldsymbol{s}, y)$$
$$\propto (1 - q) \cdot (-\log p_i) + q \cdot (1 - p_i)$$

is a weighted average of CE and RCE (MAE), where $a > 0$, $b > 0$, and $0 < q < 1$.

**Taylor Cross Entropy** (TCE; Feng et al. 2020)

$$L_{\text{TCE}}(\boldsymbol{s}, y) = \sum_{i=1}^{q} \frac{(1 - p_y)^i}{i}$$

is derived from Taylor series of the log function. It reduces to MAE when $q = 1$. Interestingly, the summand of TCE $(1 - p_y)^i / i$ with $i > 2$ is proportional to AUL with $a = 1$ and $q = i$. Thus TCE can be viewed as a combination of symmetric and asymmetric loss functions.

**Active-Passive Loss** (APL; Ma et al. 2020)

Ma et al. (2020) propose weighted combinations of active and passive loss functions. We include NCE+MAE as an example:

$$L_{\text{NCE+MAE}}(\boldsymbol{s}, y) = a \cdot L_{\text{NCE}}(\boldsymbol{s}, y) + b \cdot L_{\text{MAE}}(\boldsymbol{s}, y)$$

$$\propto (1 - q) \cdot \frac{-\log p_y}{\sum_{i=1}^{k} -\log p_i} + q \cdot (1 - p_y)$$

where $a > 0$, $b > 0$, and $0 < q < 1$.

### A.5 LOSS FUNCTIONS WITH ADDITIONAL REGULARIZERS

We additionally review loss functions that implicitly involve a regularizer and a primary loss function following the standard form Eq. (6). See Table 7 for a summary. In addition to the sample-weighting curriculums implicitly defined by the primary loss function, the additional regularizer complicates the analysis of the training dynamics. We leave investigations on how these regularizers affect noise robustness for future work.

| Name | Original | Primary Loss | Regularizer |
|------|----------|--------------|-------------|
| MSE | $1 - 2p_y + \sum_{i=1}^{k} p_i^2$ | $1 - p_y$ | $\sum_{i=1}^{k} p_i^2$ |
| PL(CR) | $-\log p_y + \log p_{y_n \mid \boldsymbol{x}_m}$ | $-\log p_y$ | $\sum_{i=1}^{k} P(\tilde{y} = i) \log p_i$ |
| CE+GLS | $-\sum_{i=1}^{k}[\mathbb{I}(i = y)(1 - \alpha) + \frac{\alpha}{k}] \log p_i$ | $-\log p_y$ | $\pm \sum_{i=1}^{k} \frac{1}{k} \log p_i$ |
| NCE | $-\log p_y / (\sum_{i=1}^{k} -\log p_i)$ | $-\gamma_{\mathrm{NCE}} \cdot \log p_i$ | $\sum_{i=1}^{k} \frac{1}{k} \log p_i$ |

Table 7: Original expressions, primary loss functions in the standard form Eq. (6) and regularizers for loss functions reviewed in Appendix A.5. We view PL in its expectation to derive its regularizer. $p_{y_n \mid \boldsymbol{x}_m}$ is the softmax probability of a random label $y_n$ with a random input $\boldsymbol{x}_m$ sampled from the noisy data. $\gamma_{\mathrm{NCE}} = 1/(\sum_{i=1}^{k} -\log p_i)$ is a scalar wrapped with the stop-gradient operator.

**Mean Square Error** (MSE; Ghosh et al. 2017)

$$L_{\mathrm{MSE}}(\boldsymbol{s}, y) = \sum_{i=1}^{k} (\mathbb{I}(i = y) - p_i)^2 = 1 - 2p_y + \sum_{i=1}^{k} p_i^2$$

$$\propto 1 - p_y + \frac{1}{2} \cdot \sum_{i=1}^{k} p_i^2 = L_{\mathrm{MAE}}(\boldsymbol{s}, y) + \alpha \cdot R_{\mathrm{MSE}}(\boldsymbol{s})$$

is more robust than CE (Ghosh et al., 2017), where $\alpha = 0.5$ and the regularizer

$$R_{\mathrm{MSE}}(\boldsymbol{s}) = \sum_{i=1}^{k} p_i^2 \tag{10}$$

increases the entropy of the softmax output. We can generalize $\alpha$ to a hyperparamter, making MSE a combination of MAE and an entropy regularizer $R_{\mathrm{MSE}}$.

**Peer Loss** (PL; Liu & Guo 2020)

$$L_{\mathrm{PL}}(\boldsymbol{s}, y) = L(\boldsymbol{s}, y) - L(\boldsymbol{s}_n, y_m)$$

makes a generic loss function $L(\boldsymbol{s}, y)$ robust against label noise, where $\boldsymbol{s}_n$ denotes the score of an input $\boldsymbol{x}_n$ and $y_m$ a label, both randomly sampled from the noisy data. Its noise robustness is theoretically established for binary classification and extended to multi-class setting (Liu & Guo, 2020).

**Confidence Regularizer** (CR; Cheng et al. 2021)

$$R_{\mathrm{CR}}(\boldsymbol{s}) = -\mathbb{E}_{\tilde{y}}[L(\boldsymbol{s}, \tilde{y})]$$

is shown (Cheng et al., 2021) to be the regularizer induced by PL in expectation. Substituting $L$ with cross entropy leads to

$$R_{\mathrm{CR}}(\boldsymbol{s}) = -\mathbb{E}_{\tilde{y}}[-\log p_{\tilde{y}}] = \sum_{i=1}^{k} P(\tilde{y} = i) \log p_i \tag{11}$$

Minimizing $R_{\mathrm{CR}}(\boldsymbol{s})$ thus makes the softmax output distribution $\boldsymbol{p}$ deviate from the prior label distribution of the noisy dataset $P(\tilde{y} = i)$, reducing the entropy of the softmax output.

**Generalized Label Smoothing** (GLS; Wei et al. 2021)

Lukasik et al. (2020) show that label smoothing (LS; Szegedy et al. 2016) can mitigate overfitting with label noise, which is later extended to GSL. Cross entropy with GLS is

$$L_{\mathrm{CE+GLS}}(\boldsymbol{s}, y) = \sum_{i=1}^{k} -[\mathbb{I}(i = y)(1 - \alpha) + \frac{\alpha}{k}] \log p_i$$

$$= -(1 - \alpha) \log p_y - \alpha \cdot \frac{1}{k} \sum_{i=1}^{k} \log p_i$$

$$\propto -\log p_y - \frac{\alpha}{1 - \alpha} \cdot \frac{1}{k} \sum_{i=1}^{k} \log p_i = L_{\mathrm{CE}}(\boldsymbol{s}, y) + \alpha' \cdot R_{\mathrm{GLS}}(\boldsymbol{s})$$

where $\alpha' = \alpha/(1-\alpha)$, has regularizer $R_{\mathrm{GLS}}$

$$R_{\mathrm{GLS}}(\boldsymbol{s}) = -\sum_{i=1}^{k} \frac{1}{k} \log p_i \tag{12}$$

With $\alpha' > 0$, $R_{\mathrm{GLS}}$ corresponds to the original label smoothing, which increases the entropy of softmax outputs. In contrast, $\alpha' < 0$ corresponding to negative label smoothing (Wei et al., 2021), which decreases the output entropy similar to $R_{\mathrm{CR}}$.

### A.5.1 DERIVATIONS FOR NCE

**Deriving Eq. (7)**  With equivalent derivatives, since

$$
\nabla_{\boldsymbol{s}} L_{\mathrm{NCE}}(\boldsymbol{s},y) = \frac{\nabla_{\boldsymbol{s}} L_{\mathrm{CE}}(\boldsymbol{s},y) \cdot \left[\sum_{i=1}^{k} L_{\mathrm{CE}}(\boldsymbol{s},i)\right] - \nabla_{\boldsymbol{s}} \left[\sum_{i=1}^{k} L_{\mathrm{CE}}(\boldsymbol{s},i)\right] \cdot L_{\mathrm{CE}}(\boldsymbol{s},y)}{\left[\sum_{i=1}^{k} L_{\mathrm{CE}}(\boldsymbol{s},i)\right]^2}
$$

$$
= \frac{1}{\sum_{i=1}^{k} L_{\mathrm{CE}}(\boldsymbol{s},i)} \left\{ \nabla_{\boldsymbol{s}} L_{\mathrm{CE}}(\boldsymbol{s},y) + \frac{k L_{\mathrm{CE}}(\boldsymbol{s},y)}{\sum_{i=1}^{k} L_{\mathrm{CE}}(\boldsymbol{s},i)} \cdot \nabla_{\boldsymbol{s}} \left[ \sum_{i=1}^{k} -\frac{1}{k} L_{\mathrm{CE}}(\boldsymbol{s},i) \right] \right\}
$$

$$
= \gamma_{\mathrm{NCE}} \cdot \left[ \nabla_{\boldsymbol{s}} L_{\mathrm{CE}}(\boldsymbol{s},y) + \epsilon_{\mathrm{NCE}} \cdot \nabla_{\boldsymbol{s}} R_{\mathrm{NCE}}(\boldsymbol{s}) \right],
$$

NCE can be rewritten as

$$L_{\mathrm{NCE}}(\boldsymbol{s},y) = \gamma_{\mathrm{NCE}} \cdot L_{\mathrm{CE}}(\boldsymbol{s},y) + \gamma_{\mathrm{NCE}} \cdot \epsilon_{\mathrm{NCE}} \cdot R_{\mathrm{NCE}}(\boldsymbol{s})$$

where $\gamma_{\mathrm{NCE}} = 1/(\sum_{i=1}^{k} -\log p_i)$ and $\epsilon_{\mathrm{NCE}} = k(-\log p_y)/(\sum_{i=1}^{k} -\log p_i)$ are scalar weights wrapped with the stop-gradient operator as discussed in §3.1. The regularizer

$$R_{\mathrm{NCE}}(\boldsymbol{s}) = \sum_{i=1}^{k} \frac{1}{k} \log p_i$$

has a form similar to $R_{\mathrm{GLS}}$.

**Deriving $\hat{w}_{\mathrm{NCE}}$ of Eq. (8)**  Here we derive the upperbound of $\|\nabla_{\boldsymbol{s}} L_{\mathrm{NCE}}(\boldsymbol{s},y)\|_1$ discussed in §3.1:

$$\|\nabla_{\boldsymbol{s}} L_{\mathrm{NCE}}(\boldsymbol{s},y)\|_1 \leq \gamma_{\mathrm{NCE}} \cdot (\|\nabla_{\boldsymbol{s}} L_{\mathrm{CE}}(\boldsymbol{s},y)\|_1 + \epsilon_{\mathrm{NCE}} \cdot \|\nabla_{\boldsymbol{s}} R_{\mathrm{NCE}}(\boldsymbol{s})\|_1)$$

$$\leq \gamma_{\mathrm{NCE}} \cdot \left( \|\nabla_{\boldsymbol{s}} L_{\mathrm{CE}}(\boldsymbol{s},y)\|_1 + \epsilon_{\mathrm{NCE}} \cdot \frac{1}{k} \sum_{i=1}^{k} \|\nabla_{\boldsymbol{s}} L_{\mathrm{CE}}(\boldsymbol{s},i)\|_1 \right)$$

$$= \gamma_{\mathrm{NCE}} \cdot \left( w_{\mathrm{CE}} \cdot \|\nabla_{\boldsymbol{s}} \Delta_y\|_1 + \epsilon_{\mathrm{NCE}} \cdot \frac{1}{k} \sum_{i=1}^{k} w_{\mathrm{CE}} \cdot \|\nabla_{\boldsymbol{s}} \Delta_i\|_1 \right)$$

$$= 2\gamma_{\mathrm{NCE}} \cdot w_{\mathrm{CE}} (1 + \epsilon_{\mathrm{NCE}})$$

$$= \hat{w}_{\mathrm{NCE}}$$

The derivation is based on the inequality $|x \pm y| \leq |x| + |y|$ and the fact that $\|\nabla_{\boldsymbol{s}} \Delta_i\|_1 = 2$. The latter can be proved by straightforward calculations. Given

$$\frac{\partial \Delta_i}{\partial s_j} = \begin{cases} 1, & j = i \\ -\frac{e^{s_j}}{\sum_{k \neq i} e^{s_k}} = -\frac{p_j}{1 - p_i}, & j \neq i \end{cases}$$

we then have

$$\|\nabla_{\boldsymbol{s}} \Delta_i\|_1 = \sum_{j} |\frac{\partial \Delta_i}{\partial s_j}| = 1 + \sum_{j \neq i} \frac{p_j}{1 - p_i} = 1 + 1 = 2$$

## B DETAILED EXPERIMENTAL SETTINGS

**Label noise**  The synthetic noisy labels are generated following (Ma et al., 2020; Zhou et al., 2021b; Patrini et al., 2017). For symmetric label noise, the training labels are randomly flipped to a different class with probabilities $\eta \in \{0.2, 0.4, 0.6, 0.8\}$. Asymmetric label noise is generated from a class-dependent flipping pattern. On CIFAR100, the 100 classes are grouped into 20 super-classes, each

|     | SCE | GCE | NCE+MAE | AUL | AGCE | AUL$^\dagger$ | AGCE$^\dagger$ |
|-----|-----|-----|---------|-----|------|------|-------|
| $a$ | /   | /   | /       | 1.1 | 0.1  | 3.0  | 1.6   |
| $q$ | 0.7 | 0.3 | 0.3     | 5.0 | 0.1  | 0.7  | 2.0   |

Table 8: Hyperparameters of different loss functions for results in §4.1 and Appendix C.1. They are tuned on CIFAR100 without label noise. Settings with inferior hyperparameters are denoted with †.

| Underfitting | Loss | CIFAR100 Acc. | $\bar{\alpha}_t^*$ | CIFAR10 Acc. | $\bar{\alpha}_t^*$ |
|--------------|------|---------------|--------------------|--------------|--------------------|
| /            | CE   | $68.76 \pm 0.21$ | 0.962 | $90.24 \pm 0.14$ | 0.624 |
| No           | SCE     | $68.89 \pm 0.05$ | 1.165 | $91.07 \pm 0.09$ | 0.726 |
|              | GCE     | $69.00 \pm 0.24$ | 0.956 | $90.83 \pm 0.20$ | 0.644 |
|              | NCE+MAE | $68.21 \pm 0.51$ | /     | $90.14 \pm 0.09$ | /     |
| Moderate     | AUL  | $47.98 \pm 3.48$ | 0.485 | $88.94 \pm 0.29$ | 0.604 |
|              | AGCE | $43.51 \pm 2.58$ | 0.406 | $90.71 \pm 0.19$ | 0.549 |
|              | NCE  | $57.95 \pm 0.26$ | /     | $85.96 \pm 0.21$ | /     |
| Severe       | MAE          | $9.11 \pm 0.83$  | 0.025 | $90.65 \pm 0.10$ | 0.355 |
|              | AUL$^\dagger$  | $10.04 \pm 2.33$ | 0.023 | $90.77 \pm 0.04$ | 0.337 |
|              | AGCE$^\dagger$ | $5.34 \pm 0.67$  | 0.008 | $81.59 \pm 8.55$ | 0.243 |

Table 9: Similar results as Table 2 with learning rate $\alpha = 0.01$. Hyperparameters for loss functions are listed in Table 8.

having 5 sub-classes. Each class is flipped within the same super-class into the next in a circular fashion. The flip probabilities are $\eta \in \{0.1, 0.2, 0.3, 0.4\}$. Human label noise for CIFAR10/100 are adopted from Wei et al. (2022). We use the "worst" labels of CIFAR10-N and the "fine" labels of CIFAR100-N, both leading to $\eta = 0.4$.

**Models and hyperparameters** We use a 4-layer CNN for MNIST, an 8-layer CNN for CIFAR10, a ResNet-34 (He et al., 2016) for CIFAR100, and a ResNet-50 (He et al., 2016) for WebVision, all with batch normalization (Ioffe & Szegedy, 2015). Data augmentation on CIFAR10/100 include random width/height shift and horizontal flip. On WebVision, we additionally include random cropping and color jittering. Without further specifications, all models are trained using SGD with momentum 0.9 and batch size 128 for 50, 120, 200 and 250 epochs on MNIST, CIFAR10, CIFAR100 and WebVision, respectively. Learning rates with cosine annealing are 0.01 on MNIST and CIFAR10, 0.1 on CIFAR100, and 0.2 on WebVision. Weight decays are $10^{-3}$ on MNIST, $10^{-4}$ on CIFAR10, $10^{-5}$ on CIFAR100 and $3 \times 10^{-5}$ on WebVision. All loss functions are normalized to have unit maximum in sample weights, which is different from (Ma et al., 2020). Hyperparameters of loss functions are listed in Tables 8 and 13 for different experiments.

## C  ADDITIONAL RESULTS TO UNDERSTAND ROBUST LOSS FUNCTIONS

We complement §4 in the main text with detailed derivations and additional results. Appendix C.1 extends discussions in §4.1 while Appendix C.2 extends §4.2

### C.1  UNDERSTANDING UNDERFITTING OF ROBUST LOSS FUNCTIONS

**Hyperparameters** We list the hyperparameters tuned on CIFAR100 without label noise in Table 8 for experiments in §4.1 and Appendix C.1.

**Robust loss functions can underfit.** In Table 9 we report results similar to Table 2 with learning rate $\alpha = 0.01$. Although settings that severe underfit slightly improve, they still perform much worse than CE, which further confirms that underfitting results from robust loss functions themselves.

**Simulated $\Delta_y$ approximate real settings.** We compare the simulated $\Delta_y$ distributions based on our assumptions in §4.1 to distributions of real datasets at initialization in Fig. 6. The expectations of simulated $\Delta_y$ follow real settings, which supports the analysis in §4.1.

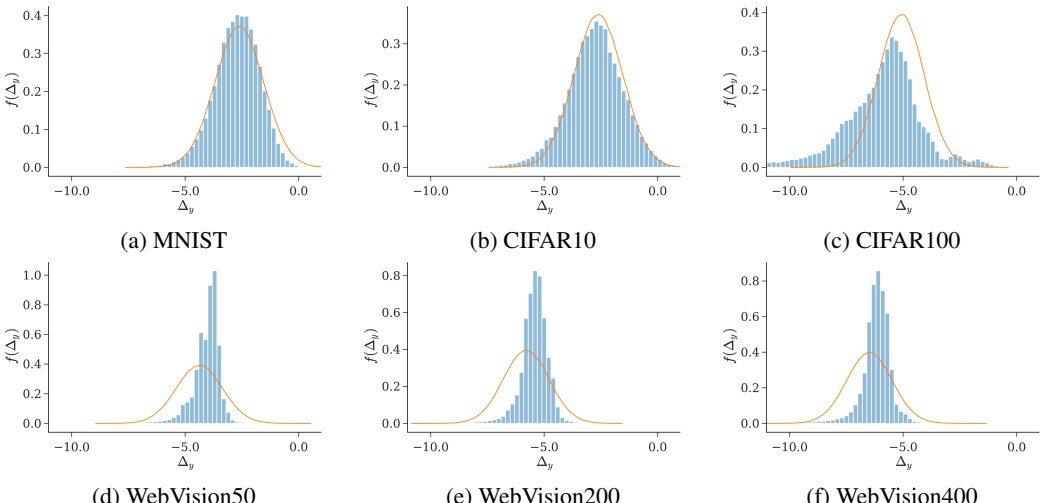

Figure 6: Comparisons between simulated and real $\Delta_y$ distributions at initialization. The simulations are based on the assumption that class scores follow normal distribution $s_i \sim \mathcal{N}(0, 1)$ at initialization and plotted as curves. Real distributions are extracted with randomly initialized models and plotted as histograms. The vertical axis denotes probability density $f(\Delta_y)$.

| | Clean | Symmetric | | | | Asymmetric | | | | Human |
| Settings | 0.0 | 0.2 | 0.4 | 0.6 | 0.8 | 0.1 | 0.2 | 0.3 | 0.4 | 0.4 |
|---|---|---|---|---|---|---|---|---|---|---|
| MAE shift | 2.0 | 2.6 | 3.0 | 4.0 | 4.0 | 2.6 | 2.6 | 3.0 | 3.0 | 2.6 |
| MAE scale | 2.0 | 2.6 | 2.6 | 3.0 | 4.0 | 2.6 | 2.6 | 3.0 | 3.0 | 2.6 |
| AGCE shift | 4.0 | 4.0 | 4.0 | 5.0 | 5.0 | 4.0 | 4.0 | 4.0 | 4.0 | 4.0 |
| AGCE scale | 2.6 | 3.0 | 4.0 | 5.0 | 5.0 | 3.0 | 3.0 | 3.0 | 3.0 | 4.0 |

Table 10: Hyperparameter $\tau$ of $w^*(\Delta_y)$ and $w^+(\Delta_y)$ for different noise settings on CIFAR100. For reference, $\mathbb{E}[\Delta_y] = 5.097$ on CIFAR100 and $\mathbb{E}[\Delta_y] = 2.717$ on CIFAR10. Better performance can be achieved with a more thorough hyperparameter search.

**Derivation of $\mathbb{E}(\Delta_y)$ at initialization in Eq. (9)**:

$$\mathbb{E}(\Delta_y) = \mathbb{E}[s_y - \log \sum_{i \neq y} e^{s_i}] = \mu - \mathbb{E}[\log \sum_{i \neq y} e^{s_i}]$$

$$\approx_1 \mu - \log \mathbb{E}[\sum_{i \neq y} e^{s_i}] + \frac{\mathbb{V}[\sum_{i \neq y} e^{s_i}]}{2\mathbb{E}[\sum_{i \neq y} e^{s_i}]^2}$$

$$=_2 \mu - \log\{(k-1)\mathbb{E}[e^{s_y}]\} + \frac{(k-1)\mathbb{V}[e^{s_y}]}{2\{(k-1)\mathbb{E}[e^{s_y}]\}^2}$$

$$=_3 \mu - \log[(k-1)e^{\mu+\sigma^2/2}] + \frac{(k-1)(e^{\sigma^2}-1)e^{2\mu+\sigma^2}}{2[(k-1)e^{\mu+\sigma^2/2}]^2}$$

$$= -\log(k-1) - \sigma^2/2 + \frac{e^{\sigma^2}-1}{2(k-1)}$$

where $\approx_1$ follows the approximation with Taylor expansion $\mathbb{E}[\log X] \approx \log \mathbb{E}[X] - \mathbb{V}[X]/(2\mathbb{E}[X]^2)$ (Teh et al., 2006), $=_2$ utilizes properties of sum of log-normal variables (Cobb et al., 2012), and $=_3$ substitutes $\mathbb{E}[e^{s_y}]$ and $\mathbb{V}[e^{s_y}]$ with expressions for log-normal distributions.

### C.1.1 ADDRESSING UNDERFITTING FROM MARGINAL INITIAL SAMPLE WEIGHTS

**Hyperparamter $\tau$ for different settings** The hyperparameter $\tau$ controlling the shape of modified sample-weighting functions $w^+(\Delta_y)$ and $w^*(\Delta_y)$ can affect the noise robustness. Thus we tune $\tau$ for the best performance under different noise types and noise rates, which are listed in Table 10.

| Loss | Clean $\eta = 0$ | Symmetric Noise (Noise Rate $\eta$) | | | |
|---|---|---|---|---|---|
| | | $\eta = 0.2$ | $\eta = 0.4$ | $\eta = 0.6$ | $\eta = 0.8$ |
| CE[‡] | $71.33 \pm 0.43$ | $56.51 \pm 0.39$ | $39.92 \pm 0.10$ | $21.39 \pm 1.17$ | $7.59 \pm 0.20$ |
| GCE[‡] | $63.09 \pm 1.39$ | $61.57 \pm 1.06$ | $56.11 \pm 1.35$ | $45.28 \pm 0.61$ | $17.42 \pm 0.06$ |
| NCE[‡] | $29.96 \pm 0.73$ | $25.27 \pm 0.32$ | $19.54 \pm 0.52$ | $13.51 \pm 0.65$ | $8.55 \pm 0.37$ |
| NCE+AUL[‡] | $68.96 \pm 0.16$ | $65.36 \pm 0.20$ | $59.25 \pm 0.23$ | $46.34 \pm 0.21$ | $23.03 \pm 0.64$ |
| AGCE | $49.27 \pm 1.03$ | $49.17 \pm 2.15$ | $47.76 \pm 1.75$ | $38.17 \pm 1.43$ | $16.03 \pm 0.59$ |
| AGCE shift | $69.39 \pm 0.84$ | $62.81 \pm 0.42$ | $48.21 \pm 1.06$ | $36.70 \pm 2.89$ | $14.49 \pm 0.17$ |
| AGCE scale | $70.57 \pm 0.62$ | $64.73 \pm 0.98$ | $56.69 \pm 0.33$ | $39.02 \pm 1.20$ | $14.64 \pm 0.79$ |
| MAE | $3.69 \pm 0.59$ | $2.92 \pm 0.46$ | $1.29 \pm 0.50$ | $2.27 \pm 1.24$ | $1.00 \pm 0.00$ |
| MAE shift | $69.02 \pm 0.78$ | $59.75 \pm 0.84$ | $44.60 \pm 0.24$ | $24.27 \pm 0.26$ | $8.08 \pm 0.26$ |
| MAE scale | $\mathbf{70.97 \pm 0.41}$ | $\mathbf{66.83 \pm 0.84}$ | $\mathbf{60.57 \pm 1.04}$ | $\mathbf{49.23 \pm 1.22}$ | $\mathbf{24.44 \pm 0.73}$ |

Table 11: Addition results to Table 3 with more symmetric label noise rates on CIFAR100.

| Loss | Clean $\eta = 0$ | Asymmetric Noise (Noise Rate $\eta$) | | | |
|---|---|---|---|---|---|
| | | $\eta = 0.1$ | $\eta = 0.2$ | $\eta = 0.3$ | $\eta = 0.4$ |
| CE[‡] | $71.33 \pm 0.43$ | $64.85 \pm 0.37$ | $58.11 \pm 0.32$ | $50.68 \pm 0.55$ | $40.17 \pm 1.31$ |
| GCE[‡] | $63.09 \pm 1.39$ | $63.01 \pm 1.01$ | $59.35 \pm 1.10$ | $53.83 \pm 0.64$ | $40.91 \pm 0.57$ |
| NCE[‡] | $29.96 \pm 0.73$ | $27.59 \pm 0.54$ | $25.75 \pm 0.50$ | $24.28 \pm 0.80$ | $20.64 \pm 0.40$ |
| NCE+AUL[‡] | $68.96 \pm 0.16$ | $66.62 \pm 0.09$ | $63.86 \pm 0.18$ | $50.38 \pm 0.32$ | $38.59 \pm 0.48$ |
| AGCE | $49.27 \pm 1.03$ | $47.53 \pm 0.73$ | $46.77 \pm 2.37$ | $39.82 \pm 2.70$ | $33.40 \pm 1.57$ |
| AGCE shift | $69.39 \pm 0.84$ | $63.03 \pm 0.42$ | $55.84 \pm 0.78$ | $49.05 \pm 0.81$ | $40.76 \pm 0.74$ |
| AGCE scale | $70.57 \pm 0.62$ | $67.13 \pm 0.60$ | $59.71 \pm 0.10$ | $48.23 \pm 0.29$ | $39.71 \pm 0.17$ |
| MAE | $3.69 \pm 0.59$ | $3.59 \pm 0.56$ | $3.19 \pm 0.98$ | $2.11 \pm 1.93$ | $2.53 \pm 1.34$ |
| MAE shift | $68.57 \pm 0.54$ | $63.44 \pm 0.32$ | $56.47 \pm 0.48$ | $48.79 \pm 1.22$ | $39.83 \pm 0.18$ |
| MAE scale | $\mathbf{70.97 \pm 0.41}$ | $\mathbf{69.50 \pm 0.24}$ | $\mathbf{64.80 \pm 0.49}$ | $\mathbf{59.04 \pm 1.52}$ | $\mathbf{44.48 \pm 1.05}$ |

Table 12: Addition results to Table 3 with more asymmetric label noise rates on CIFAR100.

**Additional results with $w^*(\Delta_y)$ and $w^+(\Delta_y)$.** We report additional results under symmetric and asymmetric label noise with diverse noise rates $\eta$ in Table 11 and Table 12, respectively. Performance of MAE and AGCE gets substantially improved with $w^*(\Delta_y)$ and $w^+(\Delta_y)$.

**Visualization of $w^*(\Delta_y)$ and $w^+(\Delta_y)$.** In Fig. 7 we visualize the shifted and scaled sample-weighting functions of MAE on CIFAR100. Although both achieve the same initial sample weights at $|\mathbb{E}[\Delta_y]|$ of CIFAR100, $w^+(\Delta_y)$ diminishes much faster as $\Delta_y$ increases, leading to insufficient learning of training samples, which can explain its inferior performance in Tables 3, 4, 11 and 12.

**Robustness of loss functions from $w^*(\Delta_y)$ and $w^+(\Delta_y)$.** Our proposed $w^*(\Delta_y)$ and $w^+(\Delta_y)$ aim to address the underfitting issue of robust loss functions with marginal initial sample weights. They modify $p_y$ into

$$p_y^* = \frac{1}{e^{-(\Delta_y/|\mathbb{E}[\Delta_y]|\cdot\tau)} + 1} = \frac{1}{e^{-(\Delta_y\cdot\alpha)} + 1}$$

and

$$p_y^* = \frac{1}{e^{-(\Delta_y+|\mathbb{E}[\Delta_y]|-\tau)} + 1} = \frac{1}{e^{-(\Delta_y+\beta)} + 1},$$

where $\alpha = \tau/|\mathbb{E}[\Delta_y]|$ and $\beta = |\mathbb{E}[\Delta_y]| - \tau$, which induces new loss functions $L^*(\boldsymbol{s}, y) = l(p_y^*)$ and $L^+(\boldsymbol{s}, y) = l(p_y^+)$, respectively. Commonly $\alpha < 1$ and $\beta > 0$ since a small $\tau$ leads to large initial sample weights and underfitting results from small $\mathbb{E}[\Delta_y]$. Notably, $\tau$ can determine the robustness of the induced loss functions. As shown in Table 10, a larger noise rate $\eta$ requires a larger $\tau$ for better performance, which assigns less weights to samples with small $\Delta_y$ in general. However, our preliminary exploration find no straightforward derivation from $L(\boldsymbol{s}, y)$ being symmetric/asymmetric to $L^*(\boldsymbol{s}, y)$ and $L^+(\boldsymbol{s}, y)$ being symmetric/asymmetric. We leave the theoretical discussions to future work.

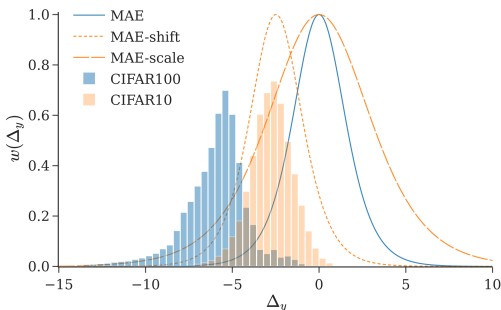

Figure 7: Shifted, scaled and the vanilla sample-weighting functions of MAE on CIFAR100. $\tau$ equals $|\mathbb{E}[\Delta_y]|$ on CIFAR10. We include the initial $\Delta_y$ distributions of CIFAR10/100 extracted with a randomly initialized model as reference.

|   | AUL | AGCE | GCE | SCE |
|---|-----|------|-----|-----|
| $a$ | 2.0 | 3.0 | / | / |
| $q$ | 2.0 | 4.0 | 0.4 | 0.95 |

Table 13: Hyperparameters of different loss functions for results in §4.2 and Appendix C.2. They are selected for broad coverage of shapes, scales and horizontal locations of sample-weighting functions instead of optimal performance on CIFAR10.

## C.2 NOISE ROBUSTNESS OF LOSS FUNCTIONS

**Computation of $\bar{w}_{\text{clean}}$ and $\bar{w}_{\text{noise}}$ for snr in Table 5** The average weight for clean samples, adjusted by the learning rate at each step $\alpha_t$, can be

$$\bar{w}_{\text{clean}} = \frac{\sum_{i,t} \alpha_t \cdot \mathbb{I}(\tilde{y}_{i,t} = y_{i,t}) w_{i,t}}{\sum_{i,t} \alpha_t \cdot \mathbb{I}(\tilde{y}_{i,t} = y_{i,t})}$$

where $w_{i,t}$ denotes the weight of $i$-th sample of the batch at step $t$, $\tilde{y}_{i,t}$ is the potentially corrupted noisy label and $y_{i,t}$ the uncorrupted label. Similarly, for noisy samples,

$$\bar{w}_{\text{noise}} = \frac{\sum_{i,t} \alpha_t \cdot \mathbb{I}(\tilde{y}_{i,t} \neq y_{i,t}) w_{i,t}}{\sum_{i,t} \alpha_t \cdot \mathbb{I}(\tilde{y}_{i,t} \neq y_{i,t})}$$

**Hyperparameters** We list the hyperparameters for different loss functions in Table 13 for results in §4.2 and Appendix C.2. In Fig. 8, we plot the sample-weighting functions of different loss functions.

**Changes of $\Delta_y$ distributions with different label noise and loss functions** Complementing Fig. 3, in Fig. 9 we plot how distributions of $\Delta_y$ change during training on CIFAR10 with additional types of label noise using hyperparameters in Table 13. They follow similar trends as in Fig. 3, thus supporting analysis in §4.2. As MAE is not robust against asymmetric label noise with high $\eta$ (Ghosh et al., 2017), it results in inferior performance. We also include results with additional loss functions in Fig. 10. Since optimal hyperparameters will result in similar sample-weighting functions, we choose hyperparameters for broad coverage of $w(\Delta_y)$ to better understand how they affect robustness.

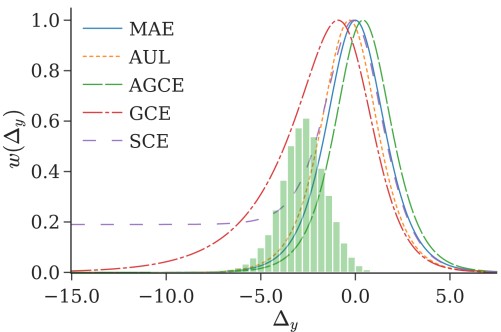

Figure 8: Plots of sample-weighting functions of loss functions used in Table 5 with hyperparameters in Table 8.

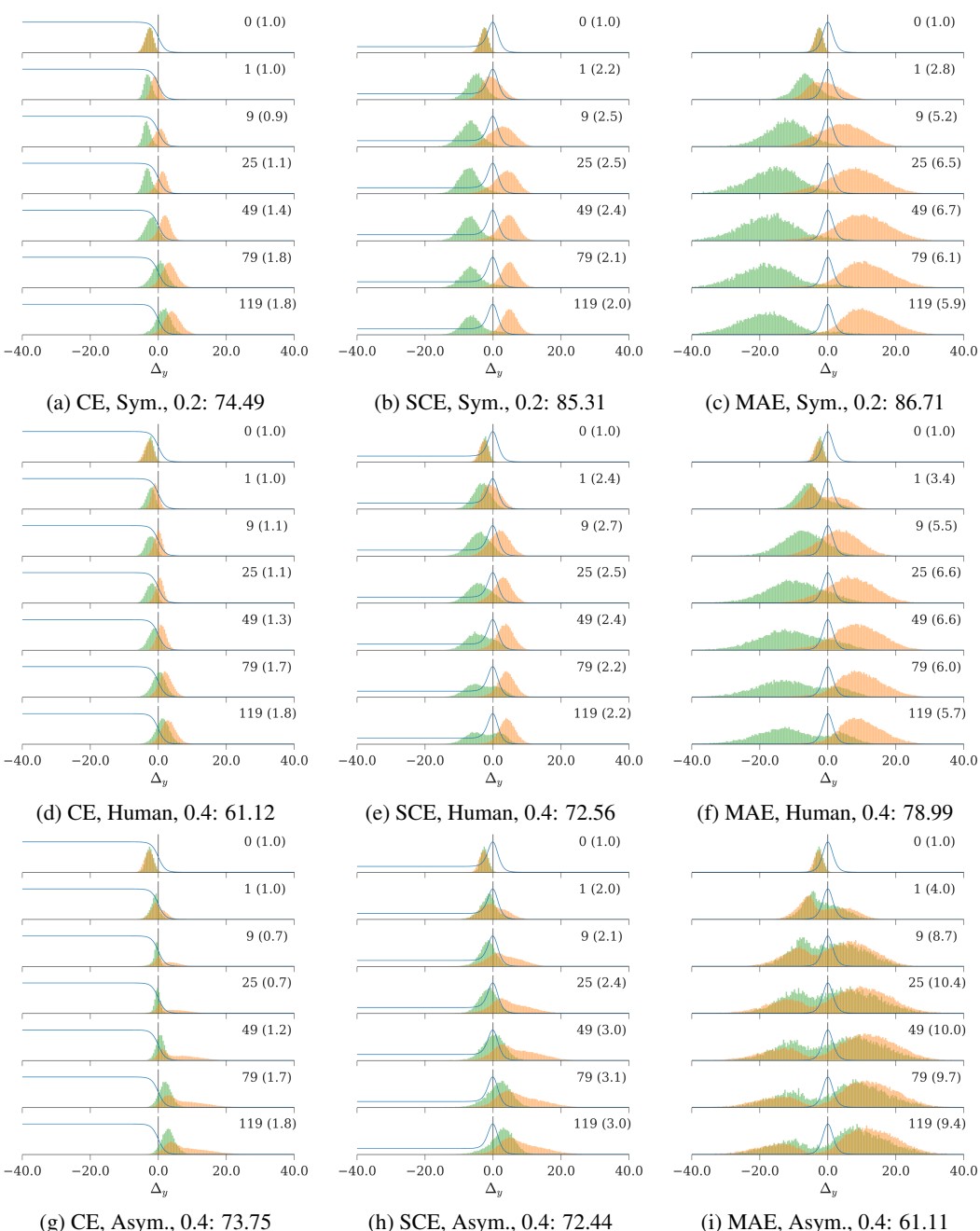

Figure 9: Additional results to Fig. 3 with different label noise: (a-c) symmetric label noise with $\eta = 0.2$; (d-f) human label noise with $\eta = 0.4$; (g-i) asymmetric label noise with $\eta = 0.4$. Noisy samples are colored green (on the left) and clean samples are orange (on the right). Test accuracies are included in the caption for reference.

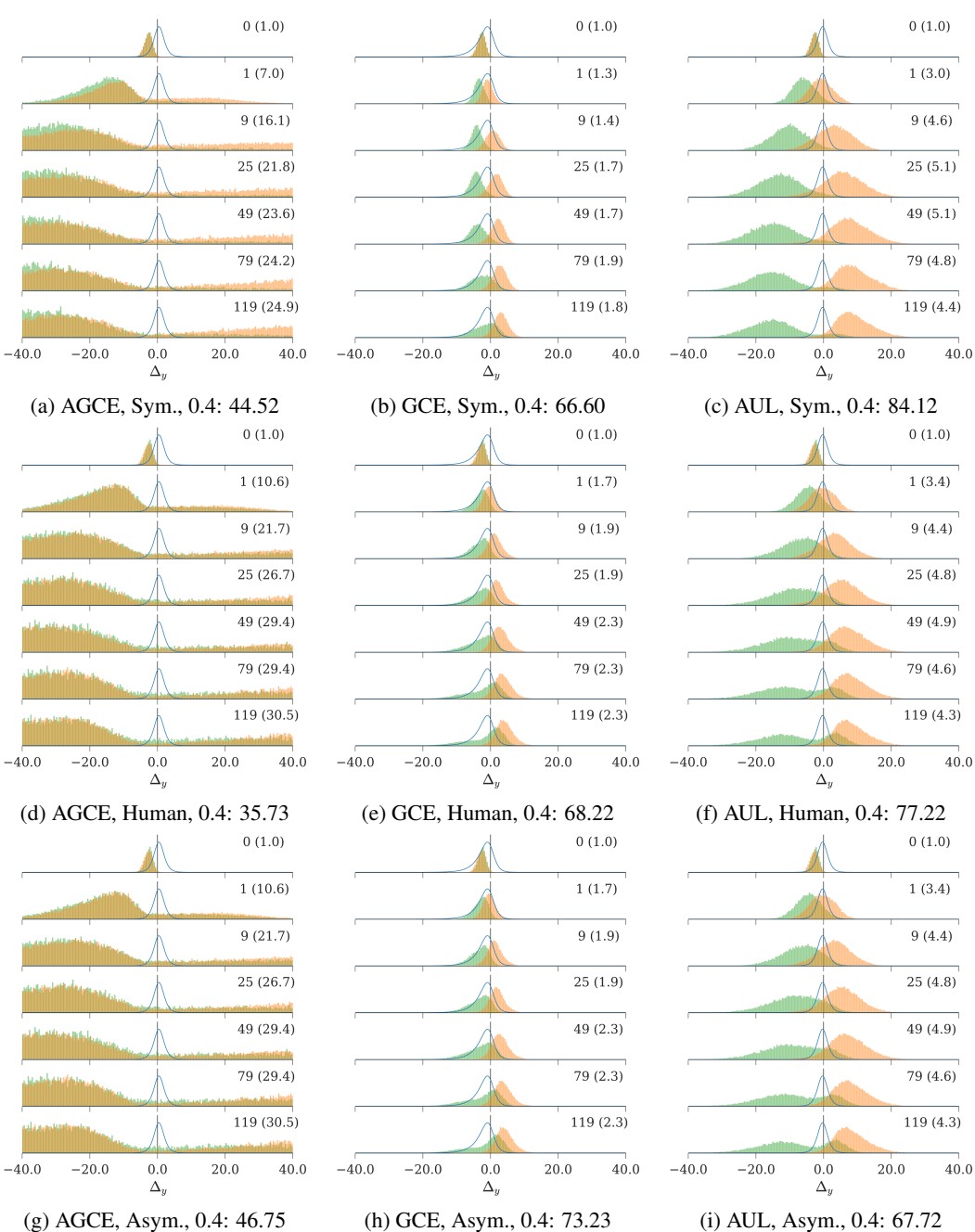

Figure 10: Additional results to Fig. 3 with more robust loss functions under different label noise: (a-c) symmetric label noise with $\eta = 0.4$; (d-f) human label noise with $\eta = 0.4$; (g-i) asymmetric label noise with $\eta = 0.4$. Test accuracies are included in the caption for reference. Noisy samples are colored green (on the left) and clean samples are orange (on the right). Hyperparameters of these loss functions are selected for broad coverage rather than optimal performance.

