# OpenReview forum: "A Curriculum Perspective to Robust Loss Functions"
_ICLR.cc/2023/Conference — Submitted to ICLR 2023_

### Official Review · Reviewer_Xqtt · 2022-10-16

**Confidence:** 4
**Correctness:** 2
**Technical Novelty And Significance:** 2
**Empirical Novelty And Significance:** 2
**Recommendation:** 3

**Clarity, Quality, Novelty And Reproducibility:**

For clarity, there are some unclear explanations at the present stage (see the above comments). For quality, although the new curriculum perspective is interesting, the quality of this paper could be further enhanced. For novelty, the technical novelty is good. However, the concerns about contributions are not addressed. For reproducibility, the experimental settings are detailed in the main paper and appendix.

**Strength And Weaknesses:**

**Strengths**

- The research problem is realistic and important. It is popular to design robust loss functions to handle noisy labels. A new perspective on the robust loss function could provide insights into the research community.
- Experimental results are detailed with extensive ablation studies.

**Weaknesses**

- The contributions of this paper are overclaimed.
- The writing and organization of this paper should be improved. It is hard to follow the implementation and claims.
- Theoretical results and discussions do not correspond to intuition and are often contradictory.

**Specification of weaknesses**

1. The paper claims that one of its main contributions is to understand why robust loss functions can underfit, and argue that prior works fail to comprehensively characterize the performance of robust loss functions. However, existing works, such as GCE, have analyzed the model robustness in training from the perspectives of gradients. It is not agnostic to training dynamics. Besides, it seems that the theoretical results of this paper are not related to a specific deep model.

2. How to understand "the weights are marginal"? Although I carefully check this paper, experimental parts, in particular, there is still no clear understanding of "marginal weights".

3. The definition of the sample weights should be given before Table 1 since $w$ appears in Table 1.

4. From Section 2.2, the paper argues that the MAE loss is passive with the first specification but active with the second. The conclusion is completely contradictory to the definitions of active and passive losses in [1].

5. The paper claims that most loss functions in Table 1 can be written as a function of the target softmax probability, which is a bit rough. From this claim, some loss functions cannot be written as this function. Therefore, theoretical results cannot be applied to them?

6. The derivation of Eq. (6) is important. More details should be provided.

7. How do Eq. (8) and the following discussions help analyze the underfitting issue? It is not clear. Besides, the theoretical results of peer loss functions are different from GCE and SCE. There is no clear explanation for relating two types of theoretical results.

8. The experimental results in Table 2 are confusing. Specifically, there are moderate underfitting for NCE and severe underfitting for MAE. Then, there is no underfitting for the combination of NCE and MAE. This is very strange. Can the underfitting issues of two loss functions be addressed by each other?

9. Eq. (9) gives the explanation that $\Delta_{y}$ is related to the class number $k$. However, it is still unclear how $\Delta_{y}$ causes underfitting.

10. Both "scale" and "shift" methods make the initial sample weights agnostic to the number of classes. From Table 4, there is a large gap between the performance of the two methods. Could the paper provide more discussions about the results? Besides, it seems that MAE totally fails in this case.

11. Does extended learning mean training with a fixed learning rate?

12. There is no clear understanding of why slowing down the learning pace can improve the robustness of a loss function. Intuitively, a slow learning pace still makes the model overfitting to mislabeled data.

----
[1] Xingjun Ma et al. Normalized loss functions for deep learning with noisy labels. ICML 2020.

**Summary Of The Paper:**

This paper presents work on learning with noisy labels. Interestingly, the paper provides a new curriculum perspective based on robust loss functions. The perspective aims to explain (1) why robust loss functions can underfit; (2) why loss functions deviating from theoretical robustness conditions can appear robust. Theoretical analyses and empirical evaluations are provided to justify the paper's claims.

**Summary Of The Review:**

This paper provides a new perspective on learning with noisy labels. However, as mentioned, there are still many unclear explanations in the present version. Therefore, before rebuttal, the reviewer is negative about this paper.

---

> ### Author Response · Authors · 2022-11-08
> **Reponse to specification of weaknesses 1-6**
>
> Your comments really help us improve the submission. We have carefully revised our submission with the details listed below. We split the responses into two halves for better readability.
>
> See the previous split for responses to weaknesses 7-12.
>
>
> ## Regarding weakness 1-6:
> > 1. The paper claims that one of its main contributions is to understand why robust loss functions can underfit, and argue that prior works fail to comprehensively characterize the performance of robust loss functions. However, existing works, such as GCE, have analyzed the model robustness in training from the perspectives of gradients. It is not agnostic to training dynamics. Besides, it seems that the theoretical results of this paper are not related to a specific deep model.
>
> The open questions we elaborated show that prior work fails to comprehensively characterize the performance of robust loss functions.
> Analysis in GCE cannot explain why NCE underfits. Besides, by factoring out the direction term with constant L1 gradient norm, MAE actually has a bell-shaped weight function as in Fig. 1(a) rather than a flat weight function suggested by the GCE paper.
>
> We added a footnote on page 1 to clarify the suitcase word "training dynamics" as "changes of model states during training except for trivial metrics like evaluation metrics and loss functions", i.e., the process to approach the risk minimizers. The exceptions are included to avoid trivialness, otherwise all research tracking the training loss analyzes "training dynamics". From this perspective, the gradient derivation in the GCE paper does not involve dynamics -- it is actually a "static" analysis. Nevertheless, we changed "Existing work ..." to "Most existing work ..." for modesty in our revision. We would be grateful for reminders of missing references.
>
> Being agnostic to a specific deep model while characterizing the performance of robust loss functions can be appealing as we discussed in the first paragraph in the Introduction. We did not argue against it.
>
> > 2. How to understand "the weights are marginal"? Although I carefully check this paper, experimental parts, in particular, there is still no clear understanding of "marginal weights".
>
> By "marginal", we mean "very small in amount or effect" (cambridge dictionary) and  "If you describe something as marginal, you mean that it is small or not very important" (collins dictionary). For better clarity, we add a footnote "Very small or too small to be important." at page 4.
>
> > 3. The definition of the sample weights should be given before Table 1 since $w$ appears in Table 1.
>
> We have aranged them accorrdingly.
>
> > 4. From Section 2.2, the paper argues that the MAE loss is passive with the first specification but active with the second. The conclusion is completely contradictory to the definitions of active and passive losses in [1]. ([1] Xingjun Ma et al. Normalized loss functions for deep learning with noisy labels. ICML 2020.)
>
> The definition of active and passive loss functions depends on specifications of the decomposition $L(s, y) = \sum_{i=1}^k l(s, i)$. In [1], there is no canonical guideline to specify the decomposition, and thus ambiguities can arise due to different specifications as we discuss in Section 2.2:
>
> Since $L_{MAE}(s, y) \propto \sum_{i=1}^k |\mathbb{I}(i=y) - p_i| \propto \sum_{i=1}^k \mathbb{I}(i=y) (1-p_i)$,
>
>   -  $l(s, i)=|\mathbb{I}(i=y) - p_i|$ satisfies the definition of passive loss $\exists i \neq y, l(s, i) \neq  0$
>
>   -  $l(s, i)=\mathbb{I}(i=y) (1-p_i)$ satisfies the definition of active loss $\forall i \neq y, l(s, i) =  0$
>
>   We made it clearer in the revised section 2. We would be grateful if you can verify the reasoning above.
>
> > 5. The paper claims that most loss functions in Table 1 can be written as a function of the target softmax probability, which is a bit rough. From this claim, some loss functions cannot be written as this function. Therefore, theoretical results cannot be applied to them?
>
> We rephrased it to "Loss functions in Table 1 except NCE and NCE+MAE can be ..." in our revision. As stated in section 3, loss functions in the form $L(s, y) = l(p_y)$ directly follow our analysis. Since NCE involves an additional regularizer, we have included extra theoretical derivations in section 3.1. We have also discussed related loss functions in appendix A.5.
>
> > 6. The derivation of Eq. (6) is important. More details should be provided.
>
> We agree with your suggestion and have included the derivation in the revised Eq. (6):
>
> $L(\boldsymbol{s}, y) = l(p_y) = \int_{\boldsymbol{s}} \nabla_{\boldsymbol{s}} l(p_{y}) \mathrm{d} \boldsymbol{s} = \int_{\boldsymbol{s}} l'(p_y) p_y'(\Delta_y) \cdot \nabla_{\boldsymbol{s}}\Delta_{y} \mathrm{d} \boldsymbol{s} =  -w(\Delta_y) \cdot \Delta_y$
>
> where $w(\Delta_y) = |l'(p_y) p_y'(\Delta_y)|$. The negative sign is due to $p_y'(\Delta_y)$.

---

> ### Author Response · Authors · 2022-11-11
> **Reponse to specification of weaknesses 7-12**
>
> Your comments really help us improve the submission. We have carefully revised our submission with the details listed below. We split the responses into two halves for better readability.
>
> See the next split for responses to weaknesses 1-6.
>
> ## Regarding weknesses 7-12
> > 7. How do Eq. (8) and the following discussions help analyze the underfitting issue? It is not clear. Besides, the theoretical results of peer loss functions are different from GCE and SCE. There is no clear explanation for relating two types of theoretical results.
>
> We changed "which helps analyze the underfitting issue in §4.1." to "Examining how $\hat w_{NCE}$ changes during training helps understand why NCE underfits in §4.1" right before section 4 for clarity.
>
> We are not sure to what "theoretical results" you refer. The connection between theoretical bounds of GCE&SCE and peer loss is orthogonal to our submission. As discussed in Appendix A, the connection from our curriculum perspective is that they all have a sample-weighting curriculum, while peer loss additionally involves a regularizer whose effects are left to future work. We have stressed this in our revision.
>
> > 8. The experimental results in Table 2 are confusing. Specifically, there are moderate underfitting for NCE and severe underfitting for MAE. Then, there is no underfitting for the combination of NCE and MAE. This is very strange. Can the underfitting issues of two loss functions be addressed by each other?
>
> Based on discussions in section 4.1, as $\hat w_{NCE}$ peaks at initialization but quickly diminishes while $w_{MAE}$ is marginal at initialization but peaks later during training, combining NCE with MAE can increase the average sample weights during different stages of training, thus mitigating the underfitting issues of each other.
>
> The discussion was originally omitted due to space limitation. We add them back to the revised section 4.1 in the paragraph "Loss combination can mitigate underfitting".
>
>
> > 9. Eq. (9) gives the explanation that $\Delta_y$ is related to the class number $k$. However, it is still unclear how $\Delta_y$ causes underfitting.
>
> Essentially, "large k" → "small initial $\Delta_y$", "small $\Delta_y$" + "sample weighting functions with marginal weights to small $\Delta_y$" → "marginal initial sample weights" → "underfitting".
>
> In the original submmision, paragraph "Underfitting from marginal initial sample weights" in section 4.1 argues that marginal initial sample weights can cause underfitting. We then explicitly discussed with: "A large k results in small ∆y at initialization, leading to marginal sample weights with the fixed sample-weighting curriculums of robust loss functions that underfit CIFAR100." right before section 4.1.1.
>
> We revised discussions right before section 4.1.1 for better clarity.
>
> > 10. Both "scale" and "shift" methods make the initial sample weights agnostic to the number of classes. From Table 4, there is a large gap between the performance of the two methods. Could the paper provide more discussions about the results? Besides, it seems that MAE totally fails in this case.
>
> Intuitively, "shift" makes weights of samples diminish much faster than "scale" with increased ∆y, which can lead to insufficient training of clean samples and thus result in inferior performance.
>
> We added the related discussions in the revised section 4.1.1. For intuitive understanding we also ploted both "shifted" and "scaled" sample-weighting functions in Fig. 7 in the appendix.
>
> And yes, MAE fails to learn on Webvision. Similar failures have been reported on CIFAR100 as we discussed in section 4.1, leading to the existing intuition: "robust loss function can underfit difficult tasks".
>
> > 11. Does extended learning mean training with a fixed learning rate?
>
> We use "extended training" for "enough training steps without early stopping or diminishing learning rates for a small training loss".
>
> We added a footnote at page 8 in our revision.
>
> > 12. There is no clear understanding of why slowing down the learning pace can improve the robustness of a loss function. Intuitively, a slow learning pace still makes the model overfitting to mislabeled data.
>
> In section 4.2.1, we argued that models start overfitting noisy samples when clean samples get well learned and have diminishing sample weights. Preventing such overfitting can thus improve the robustness.
>
> Essentially, "slow down the learning pace" → "prevent clean samples from being overly learned" → "weights of clean samples do not diminish" → "expected gradient is not dominated by noisy samples" → "noisy samples are not overfitted" → "improved robustness".
>
> We made it explicit in our revised section 4.2.1.

---

### Official Review · Reviewer_BHcb · 2022-10-24

**Confidence:** 3
**Correctness:** 3
**Technical Novelty And Significance:** 3
**Empirical Novelty And Significance:** 2
**Recommendation:** 6

**Clarity, Quality, Novelty And Reproducibility:**

- This paper is well-motivated and organized.
- The explanation of why robut losses suffer from underfitting is interesting but the performance of adjusted loss is not competitive.
- The paper does not provide the code currently.

**Strength And Weaknesses:**

**Strength:**

- By writting the losses into the form of Equation (6) is intersting, which enables us to examine the effectiveness of each loss by observing the dynamics of  $w(\delta y)$ and $\delta y$. The explanation is supported by the extensive experiments.

- The proposed shifted and scaled MAE significantly improve the performance comapred to vanilla MAE.

**Weakness:**

- Even though the explanation and claims are interesting. The performance of adjusted loss is not competetive to SOTA. For example, the performance in this paper is even not stronger than [1] which also improves the robust losses.

- [2] also observes that CE with adjusted learning rate can improve performance. Authors should cite and discuss the difference with [2].


[1] Normalized loss functions for deep learning with noisy labels

[2] O2U-Net: A Simple Noisy Label Detection Approach for Deep Neural Networks

**Summary Of The Paper:**

This paper studies why robust loss in the literature of learning with noisy labels may suffer from underfitting. Even though some early papers have already studied this question, authors provide a different perspective by showing that most robust loss functions differ only in the sample-weighting curriculums they implicitly define. Inspired this observation, authors explain the under-fitting phenomenon from the weighting term and propose an adjustment of weighting funciton to improve the model performance. Experiments are conducted on CIFAR-10, CIFAR-100 and Webvision datasets.

**Summary Of The Review:**

- The paper has merits on explaining the underfitting phenomenon of certain robust losses in the literature of learning with noisy labels. Even though
the performance is somehow limited, I think the claims and experiments may inspire further research in this area. Thus, I initially give the score of 6.

---

> ### Author Response · Authors · 2022-11-07
> **Thanks for your useful comments!**
>
> Thanks for pointing out the missing reference! In addition, we have attached our code in the revision.
>
> ### Regarding Weakness:
>
> > Even though the explanation and claims are interesting. The performance of adjusted loss is not competetive to SOTA. For example, the performance in this paper is even not stronger than [1] which also improves the robust losses.
> >
> > [1] Normalized loss functions for deep learning with noisy labels
>
> Actually our results are competitive with those of [1]. Here we list the results of our MAE-scale under different settings and the best number reported in [1]. Higher average accuracy with each setting is **bolded**. For most settings, our approach outperforms the best of [1].
>
> | Noise/Data                 | MAE-scale        | best of [1]      |
> | -------------------------- | ---------------- | ---------------- |
> | cifar100, clean            | **70.97 ± 0.41** | 69.02 ± 0.11     |
> | cifar100, sym,  $\eta=0.2$ | **66.83 ± 0.84** | 65.31 ± 0.07     |
> | cifar100, sym,  $\eta=0.4$ | **60.57 ± 1.04** | 59.48 ± 0.56     |
> | cifar100, sym,  $\eta=0.6$ | **49.23 ± 1.22** | 48.06 ± 0.34     |
> | cifar100, sym,  $\eta=0.8$ | 24.44 ± 0.73     | **25.80 ± 1.12** |
> | cifar100, asym, $\eta=0.1$ | **69.50 ± 0.24** | 66.13 ± 0.31     |
> | cifar100, asym, $\eta=0.2$ | **64.80 ± 0.49** | 63.12 ± 0.41     |
> | cifar100, asym, $\eta=0.3$ | **59.04 ± 1.52** | 58.02 ± 0.48     |
> | cifar100, asym, $\eta=0.4$ | 44.48 ± 1.05     | **47.22 ± 0.30** |
> | webvision-50               | **66.72**        | 62.64            |
>
> We also want to emphasize that we use the effectiveness of our fix for the underfitting issue to corroborate the usefulness of our curriculum perspective. Regarding this objective, the baseline should be MAE and SOTA results are only included as context.
>
>
> > [2] also observes that CE with adjusted learning rate can improve performance. Authors should cite and discuss the difference with [2].
> >
> > [2] O2U-Net: A Simple Noisy Label Detection Approach for Deep Neural Networks
>
> We add the following to the related work:
>
> "Altering noise robustness by adjusting the learning rate is reminiscent of [2]. They use a cyclic learning rate to make models change back and forth between overfitting and underfitting to collect statistics for noisy label detection. To achieve noise robustness, they discard samples with detected noisy labels and retrain the model from scratch. In contrast, our results show that simply changing the learning rate can achieve noise robustness."

---

> > ### Comment · Reviewer_BHcb · 2022-12-08
> > **Follow up**
> >
> > Thanks for the authors' reply. My concerns are addressed even though I still feel the improvements over [R1] is marginal (1% ~ 2%) and some other losses are not compared in the experimental results (e.g., Peer loss [R2], ELR [R3]). Also, it seems that authors have performed MAE scale/shift on CIFAR100 (Table 11, Table 12) but the thorough experiments on CIFAR10 are missing.
> >
> > After reading other reviews, I agree with other reviewers that there are still some unaddressed issues in the paper such as the theoretical robustness of MAE shift/scale and the experiments on other domains beyond image classification are missing.
> > But I think the part of explaining the underfitting phenomenon of certain robust losses from the perspective of weighting scheme is novel and inspiring and the authors have done great efforts to explain and analyze these losses in the main paper and supplementary material. To this end, I decide to keep my score.
> >
> >
> >
> >
> > [R1] Normalized loss functions for deep learning with noisy labels
> >
> > [R2] Peer Loss Functions: Learning from Noisy Labels without Knowing Noise Rates
> >
> > [R3] Early-Learning Regularization Prevents Memorization of Noisy Labels

---

> > > ### Author Response · Authors · 2022-12-10
> > > **Thanks for your reply!**
> > >
> > > Thanks for your effort in helping improve our submission! However, we believe there are still important misunderstandings to be clarified:
> > >
> > > First, the mention results on underfitting (section 4.1.1) serve the following reasoning:
> > > 1. Scale and shift are **effective fixes**: they significiantly improve the performance of the underfitting MAE (with a 60% increase)
> > > 2. **The results are not trivial**: they are **comparable** to the SOTA (even with an 1% ~ 2% increase)
> > >
> > > For 1, the baseline should be the underfitting loss functions to be fixed, e.g., MAE. Since the reviewed loss functions do not underfit on CIFAR10, there is no need to include shift&scaled results on CIFAR10. We will include discussions of [R3] in our appendix.
> > >
> > > For 2, the baseline can be **any** loss functions with near-SOTA performance. If we claim an overall SOTA **instead of 2**, then we should expect a nontrivial margin over the previous SOTA, and thoroughly compare our approach to other loss functions. However, we only stress 2 rather than an overall SOTA.
> > >
> > > Second, analysis following previous theoretical bounds is a "nice-to-have" for a much stronger result, but not a "must-have" in our submission focusing on the curriculum perspective.
> > >
> > > Finally, our empirical results follow the previous protocol on robust loss functions, i.e., synthetic noise on CIFAR10/100 and MNIST, and real noisy datasets like WebVision. We further include new settings on human noisy labels and WebVision beyond the subsampled 50-class settings. Extending robust loss functions to domains beyond image classification, as we have preliminarily explored, can face distinct challenges that would consistitue another submission.
> > >
> > > Please let us know if these address your concerns!

---

### Official Review · Reviewer_JUqW · 2022-10-24

**Confidence:** 3
**Correctness:** 4
**Technical Novelty And Significance:** 2
**Empirical Novelty And Significance:** 3
**Recommendation:** 6

**Clarity, Quality, Novelty And Reproducibility:**

Novelty: Good. The paper makes non-trivial advances over the current state-of-the-art.
Quality: Good. The paper appears to be technically sound. The proofs appear to be correct, but I have not carefully checked the details. The experimental evaluation is adequate and the results convincingly support the main claims.
Clarity: Good. The paper is well organized but the presentation has minor details that could be improved.
Reproducibility: Good. Key resources (e.g., proofs, code, data) are available and sufficient details (e.g., proofs, experimental setup) are described such that an expert should be able to reproduce the main results.


**Strength And Weaknesses:**

Strengths:
1. This paper explains the robustness of the loss function against label noise from a new perspective and shows most robust loss functions differ only in the sample weighting. Therefore, the robustness of some loss would be improved by adjusting the implicit sample-weighting function.
2. This paper not only analyzes why some robust losses underfit but also gives two effective solutions, scaling or shifting the sample-weighting functions.
3. Comprehensive experimental results in different cases clearly demonstrate the effectiveness of the proposed methods.
4. This paper is well-written and well-organized.

Weaknesses:
1. There is a question, that is, does robust loss (e.g. MAE) with scaled or shifted sample-weighting function still satisfy the conditions for noise robustness?
2. In nature, the methods proposed in this paper aim to adjust the sample weights. However, the authors neither compare the proposed methods with existing sample reweighting methods nor provide details and discussions for the differences between the proposed methods and existing sample reweighting methods.
3. The authors claim that robust loss functions are vulnerable to label noise with extended training and provide the result of MAE on MNIST as an example. However, such observation does not mean that all robust losses would be vulnerable to label noise with extended training. They should provide more results of other robust losses to support this statement.
4. I found some typos in this paper, so I would suggest that the authors could carefully check the manuscript repeatedly. The following are some typos I found:
- "Although the training dynamics of NCE is complicated by ..." might be "Although the training dynamics of NCE are complicated by ..." in section 3.1.
- The word "distribution" in the title of Figure 3 and Figure 5 may be in the plural.



**Summary Of The Paper:**

Learning with noisy labels is an important task in weakly supervised learning, which has received extensive attention in recent years. This paper aims to answer two questions: why robust loss functions can underfit and why loss functions deviating from theoretical robustness conditions can appear robust? Specifically, the authors rewrite the loss function into a standard form with equivalent gradients and show that most robust loss functions differ only in the sample-weighting curriculums they implicitly define. Thus, they propose to scale or shift the sample-weighting functions to address underfitting. Additionally, they also show that training schedules can affect noise robustness.


**Summary Of The Review:**

Overall, I think this paper is a relatively good work on learning with noisy labels because it provides a new perspective to explain the robustness of loss function against label noise. However, there are some typos in this paper and some details of the proposed methods are missing. So, I think the authors should carefully check the manuscript repeatedly and provide the necessary details and discussions to further polish this paper.

---

> ### Author Response · Authors · 2022-11-07
> **Thanks for your in-depth comments!**
>
> Your in-depth questions really help improve our submission! The following are the detailed responses:
>
> ### Regarding Weaknesses:
>
> > There is a question, that is, does robust loss (e.g. MAE) with scaled or shifted sample-weighting function still satisfy the conditions for noise robustness?
>
> This is a great question! However, our preliminary exploration does not find a straightforward way going from $l(p_y)$ being symmetric/asymmetric to $l(p_y')$ being symmetric/asymmetric, where $p_y' = \frac{1}{e^{-(\Delta_y + \alpha)} + 1}$  corresponds to shifted sample-weighting function while $p_y' = \frac{1}{e^{-(\Delta_y \cdot \beta)} + 1}$ to the scaled one. There could be a prove lying beyond our preliminary exploration. Our initial empirical results shows that the performance under label noise depends on the hyperparameter $\tau$ of $w^*(\Delta_y)$ and $w^+(\Delta_y)$: larger $\tau$ assigns less weights to samples with low $\Delta_y$ and leads to better performance. We added related discussions under "Robustness of loss functions from $w^*(\Delta_y)$ and $w^+(\Delta_y)$." in the revised Appendix C.1.1, while further theoretical discussions are left for future work.
>
> > In nature, the methods proposed in this paper aim to adjust the sample weights. However, the authors neither compare the proposed methods with existing sample reweighting methods nor provide details and discussions for the differences between the proposed methods and existing sample reweighting methods.
>
> We have added detailed discussions on how the implicit sample-weighting functions (A) relate to existing sample-weighting methods (B) in the related work (section 5). For better clarity we grouped all comparisons to curriculum learning into the second paragraph in the revised section 5. To summarize the differences:
>
> 1. A is implicitly defined (and we are the first to systematically identify them) while B is explicitly specified or predicted by a model.
> 2. For difficulty measure of samples, A use the more direct implicit loss function $\Delta_y$ we identified while B use measures like loss function or gradient magnitude, which can be affected by the preference from the implicit sample-weighting functions $w(\Delta_y)$ of each loss function.
> 3. A emphasizes moderately difficult samples while B emphasizes either difficult or simple samples.
>
> We think an empirical comparison to other sample reweighting methods can be orthogonal to the submission:
>
> - Our proposed scaling and shifting method aims to fix the underfitting issue of some robust loss functions, which corroborates the usefulness of our curriculum perspective.
> - We do not aim for a SOTA approach for general noise robust learning. In contrast, both methods are simple but effective fixes to the underfitting issue of robust loss functions.
>
> > The authors claim that robust loss functions are vulnerable to label noise with extended training and provide the result of MAE on MNIST as an example. However, such observation does not mean that all robust losses would be vulnerable to label noise with extended training. They should provide more results of other robust losses to support this statement.
>
> Results in section 4.2.1 only serve as exceptions to existing theoretical results on noise robustness, which corroborates the usefulness of our curriculum perspective. Due to the page limitation, we think providing an example may be enough for this purpose. In addition, we have deliberately used "can be" instead of "is/are" to emphasize "existence" rather than "universality" in the original submission. We make this point clearer in our revised version right before section 2 and in section 4.2.1.
>
> > I found some typos in this paper, so I would suggest that the authors could carefully check the manuscript repeatedly. The following are some typos I found:
> >
> > - "Although the training dynamics of NCE is complicated by ..." might be "Although the training dynamics of NCE are complicated by ..." in section 3.1.
> >
> > - The word "distribution" in the title of Figure 3 and Figure 5 may be in the plural.
>
> We have fixed the typos accordingly. In addition, we have thoroughly hunted for typos both manually and automatically.

---

> > ### Comment · Reviewer_JUqW · 2022-12-08
> > **Reply**
> >
> > Thanks for the response and sorry for the late reply. I am partially satisfied with the response. I feel it is a pity that MAE with scaled or shifted sample-weighting function may not satisfy the conditions for noise robustness. As the authors admit that further theoretical discussions might be required. Besides, since this paper involves the sample-weighting perspective, I still think that empirical comparisons against simple sample-weighting methods (and the simple small-loss trick) are required,  which needs more empirical studies. On the other hand, I acknowledge the contribution of analyzing robust loss functions from the sample-weighting curriculum perspective.
> >
> > So I would keep my score.

---

> > > ### Author Response · Authors · 2022-12-10
> > > **Thanks for you reply!**
> > >
> > > Thanks for providing such valuable feedback!
> > >
> > > Although the theoretical bound for the shifted&scaled loss functions is very interesting to explore, it is only a related topic for the current submission focussing on the curriculum perspective, which is very different from existing theoretical perspectives. Connecting results of the two perspectives is an important research topic for future work. Intuitively, if there exists a theoretical bound, it should heavily depend on the value of $\tau$.
> > >
> > > There still remain concerns about empirical comparison against simple sample-weighting methods. We think its necessity depends on what we have claimed:
> > >
> > > If we claim a novel approach to the sample-weighting method or a new overall SOTA result for noise robust training, then the comparison is a must. However, we claim neither except for a fix of the underfitting robust loss functions. In this regard, we only need to demonstrate that scale and shift are effective fixes. Specifically,
> > > 1. Our fix significantly improves underfitting robust loss functions
> > > 2. The resulting performance is nontrivial: it is comparable to SOTA robust loss functions
> > >
> > > We believe that the current results support our claim given the tight page limitation. However, we will consider adding related results in our later revisions if you still insist.
> > >
> > > Please let us know if these address your concerns!

---

### Official Review · Reviewer_N4nc · 2022-10-27

**Confidence:** 4
**Correctness:** 3
**Technical Novelty And Significance:** 4
**Empirical Novelty And Significance:** 3
**Recommendation:** 6

**Clarity, Quality, Novelty And Reproducibility:**

### Novelty
Its novelty comes from providing a novel view (supported by experimental evidences) about an existing, well-known problem. I have no big concern about novelty.

### Quality
Their experiments cover various types of loss functions and different hyperparameter settings. The experiments support the claims well, but they have to mention to what extent the claims can be generalized (as I mentioned in the Weakness).

### Clarity
Several minor improvements are needed.
- Pg4. Are those Wei et al.'s human noisy labels and WebVision's noisy dataset much closer to the symmetric or asymmetric?
- Pg5. section 4.1. It is not clear what 'Robust loss function' means and what are those. Does it indicate loss functions known to mathematically satisfy the condition (1)? Or loss functions showing a good performance in Table 2?
- Pg5. section 4.1. It is not clear what 'parameter update size' means.
- Fig 1. Can you provide more detail how you get the initial distribution of CIFAR10 and CIFAR100?

### Reproducibility
The authors said their code will be available at github, but not provided in this submission. So I cannot evaluate its reproducibility.

**Strength And Weaknesses:**

## Strength
This paper provides an interesting view for under-fitting issues in loss functions known to be robust against noisy labels. Various loss functions with various parameter settings are considered to be examine and observations are well aligned with our intuition.

## Weakness
- The experiments are mainly focusing on image classification, and on convolutional layer based models. The authors should discuss in the paper if they want to claim their findings can be generalized to any other tasks, and model architectures. Or it should clearly mention to what extent was this experiment covers in the main text. It is important because the paper's main claims are mainly supported my experimental results.
- The main manuscript is not sufficiently self-inclusive. Readers should check the supplementary material to check important experimental settings, and term definitions. I understand this is mostly because this paper have to introduce many terms, measures, and acronyms. But such a paper should be more cautious in clarity. See comments in Clarity for actionable feedbacks.

**Summary Of The Paper:**

This paper provides a novel view to examine robustness against label noise and its under-fitting issue. It first points out previous arguments could not sufficiently address causes of the under-fitting issue. In contrast with the previous approaches, it factorizes a loss function with the sample-weighting function and the distribution of implicit losses, which provides a novel view to see it as a type of curriculum learning. It provides several experimental results showing training dynamics along with several measures based on the factorized terms, and makes arguments about the under-fitting issue and robustness.

**Summary Of The Review:**

This paper provides a novel, and potentially impactful view about under-fitting problems of robust loss functions. They made several sub-claims based on their observations, which is intuitively understandable. Experiments are well designed, and the claims are sound in some extent, but they should clearly state that to what extent they want to generalize the claims. Also there are some minor issues including writing clarity and reproducibility. I am willing to raise the score upon the authors response.

---

> ### Author Response · Authors · 2022-11-07
> **Thanks for your constructive comments!**
>
> Thanks for your constructive comments! We have thoroughly hunt for ambiguities and typos. The resulting revision should have higer quality.
>
> ### Regarding Weaknesses
>
> > The experiments are mainly focusing on image classification, and on convolutional layer based models. The authors should discuss in the paper if they want to claim their findings can be generalized to any other tasks, and model architectures. Or it should clearly mention to what extent was this experiment covers in the main text. It is important because the paper's main claims are mainly supported my experimental results.
>
> We add the following discussions in the revised section 6 (conclusion and discussions):
>
> "As with previous work on robust loss functions, our empirical results are based on image classification using convolutional neural networks with and without residual connections. We have extended our experiments to cover larger-scale classification tasks, human label noise and a broader array of robust loss functions. Although our derivation does not depend on the models and task specifications, additional experiments should be performed in future work to extend our conclusions to more models and tasks."
>
> Note that we follow similar protocols as in previous work to better contrast our findings with those from previous theoretical approaches. Extending our findings to other tasks involving the cross entropy loss, e.g., text generation and object recognition, is very interesting for future work.
>
> > The main manuscript is not sufficiently self-inclusive. Readers should check the supplementary material to check important experimental settings, and term definitions. I understand this is mostly because this paper have to introduce many terms, measures, and acronyms. But such a paper should be more cautious in clarity. See comments in Clarity for actionable feedbacks.
>
> We have improved the self-inclusiveness given tight page limitation. See the following responses for details of our revision.
>
> ### Regarding Clarity
>
> > Pg4. Are those Wei et al.'s human noisy labels and WebVision's noisy dataset much closer to the symmetric or asymmetric?
>
> Human noisy labels can exhibit more complex noise patterns than symmetric or asymmetric label noise. We modify "For real world settings we also include results on CIFAR10/100 with human noisy labels (Wei et al., 2022b) and the large scale noisy dataset WebVision (Li et al., 2017)." to "For real-world scenarios, we include CIFAR10/100 with human label noise (Wei et al., 2022) and the large-scale noisy dataset WebVision (Li et al., 2017), which exhibit more complex noise patterns than symmetric and asymmetric label noise." at the beginning of the revised section 4.
>
> > Pg5. section 4.1. It is not clear what 'Robust loss function' means and what are those. Does it indicate loss functions known to mathematically satisfy the condition (1)? Or loss functions showing a good performance in Table 2?
>
> A rigorous definition of "robust loss function" is a loss function satisfying Eq (1). But similar to existing work, as a slight abuse of terminology, we use "robust loss functions" to refer to loss functions with less performance drop under label noise than cross entropy. In our revised version, we explicitly state it right before section 2.1.
>
> > Pg5. section 4.1. It is not clear what 'parameter update size' means.
>
> We rephrased it to "the effective scale of parameter update at step $t$ during SGD".
>
> > Fig 1. Can you provide more detail how you get the initial distribution of CIFAR10 and CIFAR100?
>
> We have clarified it in the caption of Fig 1. They are obtained by computing $\Delta_y$ with a randomly initialized model for all training samples.
>
> ### Regarding Reproducibility
> Please see the code in the supplementary materials of our revision.

---

### Author Response · Authors · 2022-11-09
**Discussions of potential impacts**

Although we focus on analyzing underfitting on difficult task and robustness against label noise for robust loss functions, our derived standard form of loss functions (which leads to our novel curriculum perspective), $L(\boldsymbol{s}, y) =  l(p_y) = -w(\Delta_y) \cdot \Delta_y$, can have broad impacts beyond research on robust loss functions, where $p_y$ is the softmax probability for the target label $y$, $w(\Delta_y)$ is the sample-weighting function and $\Delta_y$ is the implicit loss function:

1. It is related to the **design** and **analysis** of loss functions. For example,
    - we can explicitly design loss functions to have a desirable $w(\Delta_y)$ for problems such as class imbalance
    - we can decide whether to mix $w(\Delta_y)$ in addition to gradients of $\Delta_y$ when combining multiple loss functions in multitask learning
    - acknowledging the implicit sample weights $w(\Delta_y) = 1 - p_y$ of cross entropy can be important in many intuitive/theoretical analysis.

2. The identified $w(\Delta_y)$ and $\Delta_y$ can help design better learning curriculums.
    - by isolating the preference from $w(\Delta_y)$, $\Delta_y$ can be a better metric for sample difficulty compared to those based on loss or gradient.
    - $w(\Delta_y)$ of robust loss functions emphsizes moderately difficult samples. A further analysis of how such $w(\Delta_y)$ affect training can help resolve the ongoing debate on whether easier first or harder first is better for sample weighting in curriculum learning.

3. It provides a new perspective to analyze the training dynamics of the model:
    - the interaction between $w(\Delta_y)$ and $\Delta_y$ help understand why samples with noisy labels only get learned in the later stage of training
    - when we are only interested in the direction of gradients, we can avoid the complications of $w(\Delta_y)$ by only considering the gradients of $\Delta_y$

---

### Decision · Program_Chairs · 2023-01-20

**Decision:**

Reject

**Justification For Why Not Higher Score:**

Reviewers still have following outstanding concerns:

1. Robust loss functions (e.g., MAE) with the proposed scaled or shifted sample-weighting function may not satisfy the conditions for noise robustness, and it may require further theoretical discussions on this fix.
2. Since this paper involves the sample-weighting perspective, empirical comparisons against simple sample-weighting methods (and the simple small-loss trick) will be good to have in the paper. Can something be said about explicit sample reweighting methods?


**Justification For Why Not Lower Score:**

N/A

**Metareview: Summary, Strengths And Weaknesses:**

The paper analyzes two aspects of robust loss functions -- why robust loss functions can underfit; and why loss functions deviating from theoretical robustness conditions can appear robust. Authors show that most robust loss functions differ only in the sample-weighting curriculums they implicitly define. They propose to scale or shift the sample-weighting functions to address underfitting and empirically show that this scale/shift operation is able to improve the performance of robust losses for noisy labels. Reviewers were split on the paper with some placing the paper above the acceptance bar due to a novel perspective (implicit weighting) for analyzing robust losses and suggesting a fix with shifting/scaling the weights. However reviewers still have following outstanding concerns --

1. Robust loss functions (e.g., MAE) with the proposed scaled or shifted sample-weighting function may not satisfy the conditions for noise robustness, and it may require further theoretical discussions on this fix.
2. Since this paper involves the sample-weighting perspective, empirical comparisons against simple sample-weighting methods (and the simple small-loss trick) will be good to have in the paper. Can something be said about explicit sample reweighting methods?

Meta-reviewer thinks the paper will indeed benefit from addressing these points and it falls below the acceptance bar in current form.




**Summary Of Ac-Reviewer Meeting:**

Only one reviewer was willing to meet virtually (one reviewer preferred to leave a comment on openreview -- others didn't respond).